# Spatiotemporal computations in the insect celestial compass

Evripidis Gkanias ✉ & Barbara Webb

Obtaining a geocentric directional reference from a celestial compass requires compensation for the sun's movement during the day (relative to the observer), which depends on the earth's rotation, time of year and the observer's latitude. We examine how insects could solve this problem, assuming they have clock neurons that represent time as a sinusoidal oscillation, and taking into account the neuroanatomy of their celestial compass pathway. We show how this circuit could exploit trigonometric identities to perform the required spatiotemporal calculations. Our basic model assumes a constant change in sun azimuth (the 'hour angle'), which is recentred on solar noon for changing day lengths. In a more complete model, the time of year is represented by an oscillation with an annual period, and the latitude is estimated from the inclination of the geomagnetic field. Evaluating these models in simulated migration and foraging behaviours shows the hour angle may be sufficient.

Using the sun (or equivalent celestial cues) as a compass requires a compensation mechanism for its movement during the day. Taking inspiration from insects, engineers have implemented various forms of celestial compass for use in robot navigation. Typically, the robot's deployment has been short enough (less than an hour) for the sun's movement to be neglected[1,2]. For longer deployment, a system clock, global positioning system (GPS) and an accurate model of the solar ephemeris can be used to time-compensate the compass[3]. However, typical use cases for a robot celestial compass would be situations where GPS is unavailable (remote locations, or compromised satellite systems). Insects use their celestial compass to navigate in deserts[4,5] and across oceans[6,7], but it is implausible to suppose they have either an absolute time reference or a precise look-up table for the solar ephemeris. Here, we propose plausible neural mechanisms by which insects could use their internal sense of time to predict the sun's position, transforming the measured azimuth of the sun into a geocentric heading reference.

Internal clocks (processes that track time or external rhythms) are widespread in biology. Insects are known to have clock input to multiple circuits[8,9]. More specifically, clock inputs from antennae[10–14] or the accessory medulla of the optic lobes[15–19] (Fig. 1a) affect celestial compass function. The proteins timeless (Tim) and period (Per) are involved in the encoding of time in insects[20]. Additionally, cryptochrome (Cry) proteins act as photoreceptor inputs to the clock

mechanism, tuning the Per and Tim mRNA oscillations to light-darkness rhythms[11–13] (also type 2 Cry (Cry2) oscillations in monarch butterflies). As Fig. 1b shows, the oscillation phases of Cry2 and Tim proteins differ by six hours, suggesting that these two signals might respectively encode the sine and cosine functions of an angle changing with constant rate of 15° per hour[21] (Fig. 1c) to complete a circle every day, like the hour hand of a clock (Fig. 1d).

Inside the insect brain, time is represented by clock neurons[8,9] whose activity depends on the mRNA levels of the proteins described above (although their exact relationship is unclear). Clock neuron activity expresses an endogenous oscillation with periods that can vary from daily to annual[20]. Approximately half of the clock neurons in the insect brain rely on the blue light sensitive type 1 Cry (Cry1)[19]. The only clock neurons that intersect the celestial compass pathway of fruit flies (*Drosophila melanogaster*) before it reaches the navigation-specific circuits of central complex (CX)[16,22] are DN1pB[23]. The current assumption is that these neurons allow the celestial compass to compensate for the moving sun by predicting the sun's course during the day[23]. However, the mechanism that allows this compensation and transforms the retinotopic position of the sun into a geocentric compass is still unknown.

Some proposed mechanisms to compensate a celestial compass for the sun's movement do not require a clock. These suggest the calibration of the celestial with other compasses[24], use the sun's

School of Informatics, University of Edinburgh, EH8 9AB Edinburgh, UK. ✉e-mail: ev.gkanias@gmail.com

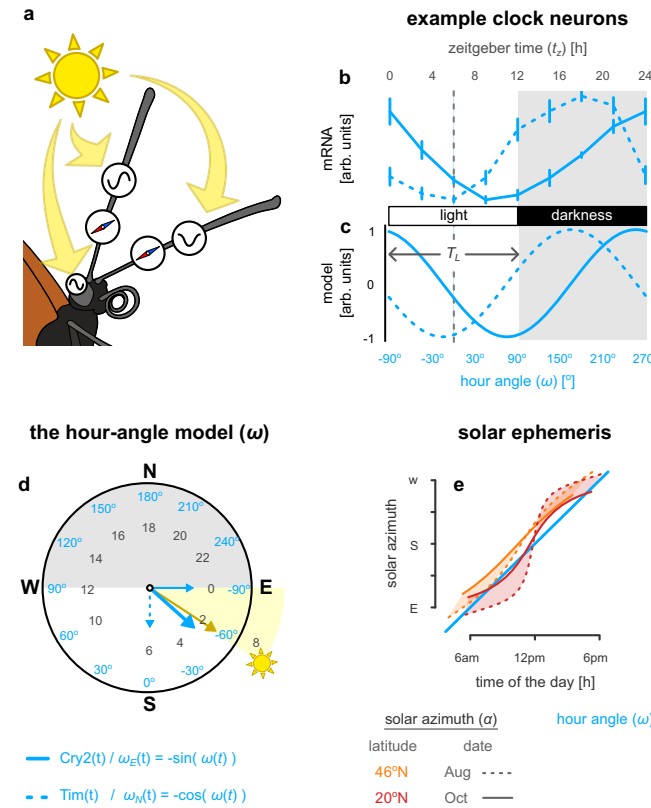

**Fig. 1 | Predicting the sun's position using light. a** Cryptochrome (Cry), timeless (Tim) and period (Per) proteins are responsible for encoding time in the insect brain. The mRNA level of Cry1 is increased in the butterfly antennas (and eyes in several insects) when they detect blue light, synchronising internal clocks to the light-and-darkness cycle. **b** The mRNA levels of Cry2 (solid line) and Tim (dashed line) proteins increase and decrease with a different phase during the 24-hour light-and-darkness cycle in monarch butterflies[11,12]. Values are the mean ± SEM (standard error of the mean) of the minimal level for each gene of $n = 4$ antennas. Data replotted in arbitrary units (arb. units) with permission from ref. [12]. **c** The mRNA levels of Cry2 and Tim proteins modelled as cosine and sine functions of zeitgeber time encode the hour angle ($\omega$). $T_L$: day length. **d** The hour angle, encoded as in **c**, is an arrow that rotates clockwise (CW) and completes a full circle daily. The arrow should point east during sunrise, south at noon and west during sunset (assuming twelve hours day and twelve hours night), roughly predicting the solar azimuth. N: north, E: east, S: south, W: west. **e** The hour-angle predicts a constant change in the solar azimuth during the day (blue line), but the actual azimuth deviates depending on location and time of year (orange and red lines). The orange lines illustrate the sun's courses in August (dashed) and October (solid) at Michigan (~46°N). The red lines show the respective sun's courses in Mexico (~20°N). Shaded areas highlight the difference in the sun's course over two months at the same location.

elevation to estimate its speed of motion[25], or assess the rotation of the skylight polarisation pattern to detect the true north[26]. Others conceptually examine the effects of mappings from different forms of clock input to the estimated sun's position, but without discussing possible biological implementations[27]. One mechanism examined in Massy et al. (2023)[27] is time-averaging, which has been linked to properties of clock proteins found in the butterfly antennas[21]. As already mentioned, the levels of these proteins express a sinusoidal pattern during the day, representing an arrow that encodes time. Schlizerman et al. (2016)[21] use this signal to estimate the sun's movement as a constant speed of 15° per hour (the speed at which the earth rotates around its axis) and assume that the clock resets at sunrise. However, this produces a directional drift in the compass reference as the length of the day (and hence the time of sunrise) changes; the clock would need to be synchronised to solar noon to provide a stable reference point. Furthermore, depending on the time of year or

location on earth, the observed movement of the sun can deviate significantly from a constant speed (Fig. 1e).

We present a neurally plausible mechanism by which a clock signal carried by DN1pB neurons could be used to correct the insect's celestial compass. We propose two alternative models to generate the DN1pB clock signal. Our hour-angle model uses the time of the day to predict the earth's rotation around its spin axis relative to the sun and adapts to changing day length. Our complete model also uses the time of year and the geomagnetic inclination (detectable by some insects[28,29]) to calculate the exact course of the sun during the day. Our results suggest that the hour-angle model is sufficient to support the behaviour of migrating and central-place foraging insects, assuming they have evolved to use an appropriate variant for the northern or southern hemisphere or the equatorial region. However, the complete model can adapt to any location, including change from clockwise (CW) to counter-clockwise (CCW) movement of the sun, and produces more accurate navigation. Some insects may require such accuracy, and the model may also have value for a localisation technology independent of GPS.

## Results

### The hour-angle model

The hour angle ($\omega$) describes the earth's rotation around its axis relative to the sun as a constant rate of 15° per hour. By convention, the hour angle is the longitudinal difference between the observer and the subsolar point (location on earth where the sun is at its zenith). It is zero when the sun is closest to the observer's zenith (solar noon), at which point the sun azimuth indicates south (assuming the observer is in the northern hemisphere). If the time of day is measured from sunrise (zeitgeber time, $t_z = t - t_{sr}$, from the German for 'time-giver'), the hour angle is given by

$$\omega(t) = \left( t_z - \frac{T_L}{2} \right) 15^{\circ},  \qquad (1)$$

where $T_L = t_{ss} - t_{sr}$ is the day length, computed as the difference between the sunrise ($t_{sr}$) and sunset ($t_{ss}$) times in hours. Supplementary Text S1 describes a possible neural implementation of equation (1), given the neural encoding of the zeitgeber time and day length separately.

Given their six-hour phase difference (equivalent to 90° hour-angle difference; see Fig. 1b), Tim and Cry2 proteins could encode the (negative) sine and cosine of the hour angle as its east-most ($\omega_E$) and north-most ($\omega_N$) components (Fig. 1c). The period of these sinusoids is determined by the temporal difference between two consecutive sunrises[30]. Their phase is determined by the day length ($T_L$), which ensures that the sinusoids are centred at solar noon (see Supplementary Fig. S1). Using both (east-most and north-most) components creates an arrow that rotates with time CW, like the hour hand of a clock (Fig. 1d, blue arrow). Negating the north-most component ($\omega_N = -\omega_N$; flipping the vertical vector in Fig. 1d) implements a CCW hour angle model, which works in the southern hemisphere. Near the equator, the sun moves almost vertically (crossing the zenith) from east to west. In this case, a zero north-most component ($\omega_N = 0$; no vertical vector in Fig. 1d) would be more accurate and avoids any assumption regarding CW or CCW sun movement. These three versions of the hour angle model allow for a reasonable approximation of the solar azimuth for different locations on earth. A complete compensation mechanism should consider more factors (Fig. 1e).

### Estimating the day length

The blue-light sensitive Cry1 proteins detected in *Danaus plexippus* monarch butterflies (equivalent to the Cry proteins of *D. melanogaster* fruit fly) synchronise the clocks to the light-dark rhythm[11,12,19]. We suggest that two kinds of synchronisation co-occur. The first

synchronises the zeitgeber time ($t_z$) with the sunrise, resetting it to zero, while the second shifts the hour angle relative to $t_z$, as described in equation (1), so that it is zero at solar noon. The required shift relative to solar noon depends on the day length ($T_L$), which we assume is a continuously updated estimate based on the changing level of skylight irradiance.

We assume the protein level $\mathrm{Cry1}(t) = a\, I_{sky}(t)$, where $a$ is a scaling factor (gain) and $I_{sky}$ is the overall blue irradiance in the sky (see Fig. 2a). We then propose that the day length ($T_L$) can be calculated as a function of Cry1,

$$\tau_L \frac{\mathrm{d}T_L}{\mathrm{d}t} = \mathrm{Cry1}(t) + \beta - T_L,\qquad(2)$$

where $\tau_L$ is a time-constant, and $\beta$ is a constant input. Parameters $a$, $\beta$, and $\tau_L$ can be optimised to fit equation (2) to the actual day length (values summarised in Supplementary Table S1). Figure 2b demonstrates the precision of this fit, where we simulate the estimated $T_L$ and plot it over the actual day length, assuming continuous light exposure during the day.

We also tested how interrupted light exposure, as might be experienced by central-place foraging insects, affected the estimates of $T_L$ (it might also affect synchronisation of $t_z$, but we do not consider that here). We simulated species with different average foraging durations, from one to eight hours per day; foraging each day could vary around this average and occur anytime from sunrise to sunset. Parameters needed to be optimised to fit the different average foraging durations (summarised in Supplementary Table S1): shorter foraging durations require longer time constants ($\tau_L$) and higher gains ($a$) to transform the sky irradiance into Cry1. Using these optimised parameters, Fig. 2c shows that the higher level of skylight irradiance measured (for the same average exposure time) during the summer (compared to winter) suffices to estimate the day length for central-place foragers. A consistent daily foraging duration and time window would result in smoother estimates. In Supplementary Text S2 and Supplementary Fig. S2, we explain how a neural filtering process could further reduce the high-frequency oscillations in this estimate.

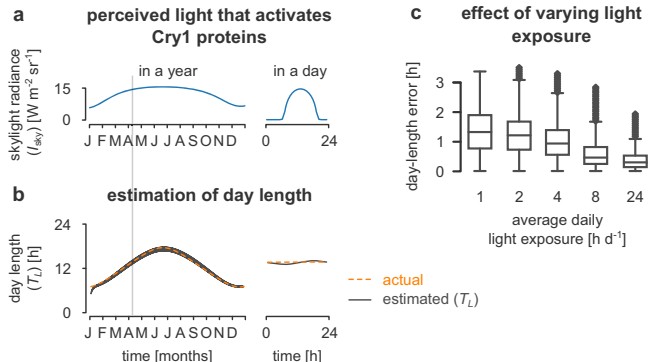

**Fig. 2 | Calculation of the day length. a** The expression of type 1 cryptochromes (Cry1) is proportional to the skylight irradiance ($I_{sky}$). On the left, we show the highest skylight irradiance for each day in a year. On the right, we show the actual irradiance in each hour on April 7 (highlighted on the left). The letters on the horizontal axis indicate the month. **b** Our estimates of the day length are updated using the Cry1 mRNA levels, resulting in an annual oscillation (black line) that follows the specific day's actual length (orange line). **c** Day length estimation is still possible (although with decreasing accuracy) as the average daily light exposure of the insect decreases to as little as one hour. This assumes the parameters for estimation can be adjusted, for example, by evolving to match a specific foraging pattern. Each box shows the quartiles of the data ($n$ = 8 784 samples span over the year 2024; 366 × 24). The whiskers extend to show the rest of the distribution except for points that exceed 1.5 × the inter-quartile range, which are marked as outliers.

## Integration with compass neurons

The calcium level of dorsal (DNs) and lateral (LNs) clock neurons in the brain of *D. melanogaster* changes over the day[19,31]. This can be described by sinusoidal functions similar to protein levels in Fig. 1b. A pair of DNs (DN1pB) target the TuBu1 neurons and provide clock information to the celestial compass pathway[8,9,32] in each hemisphere[19,23]. We hypothesise that the calcium level of DN1pB neurons predicts the solar azimuth ($\alpha$) and corresponds to the east-most ($\omega_E$) and north-most ($\omega_N$) components of the hour angle. There are other clock neurons innervating the ellipsoid (EB) and fan-shape body (FB) of the CX, but we would argue that time compensation should occur before the celestial compass integrates with other compasses (such as wind or visual landmarks) in the EPG neurons of the EB. The DN1pB neurons are, so far, the only reported neurons to target the celestial compass pathway upstream of the EB in *D. melanogaster*[19]. Although they receive limited direct input from sensory-driven clocks in the antennae or retina (see Fig. 3a and Supplementary Table S2, generated using data from the FlyWire database[33]), they receive input from these via multiple interneurons, which might have a role in shaping their response into a smooth sinusoid (Supplementary Text S2).

Previous computational modelling studies provide a good approximation of how the TuBu1 neurons can encode the solar azimuth[25,34]. In *D. melanogaster*, polarised light is detected from the dorsal rim area and travels to the anterior optic tubercle (AOTu) through the retina and the medulla (Fig. 3a)[35,36]. Colour information is detected from the remaining dorsal area in the eye. It travels through the dorsal retina and medulla to AOTu and the TuBu1 neurons, where it probably integrates with polarisation information to estimate the retinotopic solar azimuth ($\alpha'$). As described above, TuBu1 neurons are also terminals for the DN1pB neurons[8,9,23,32]. Thus, we assume that the time-based solar azimuth prediction is combined with the retinotopic solar azimuth estimate at the axons of TuBu1 neurons, transforming it into a geocentric compass in the anterior bulb.

There are two types of TuBu1 neurons: TuBu1a, which targets ER4m ring neurons, and TuBu1b, which targets both ER4m and ER5 neurons (see Supplementary Table S3). Calcium recordings of TuBu1 neurons suggest that the two hemispheres independently encode the (retinotopic) solar azimuth in a spatial sinusoidal pattern of activity[36]. For our proposed mechanism of time compensation to work, we need to assume that the TuBu1a and TuBu1b populations express a 90° shift in the represented direction of the sun (see Fig. 3b). We also assume that $DN1pB_E$ targets the TuBu1a population and the $DN1pB_N$ targets the TuBu1b population. Finally, we assume the responses of the TuBu1 populations are added together to encode the animal's heading in the responses of the ER4m population.

The above process implements a crucial trigonometric identity that transforms the solar azimuth into a geocentric compass in the ring neurons, described as

$$\begin{aligned} \mathrm{ER4m}^n &= \sin(\alpha)\cdot\sin(\alpha'-\phi^n) + \cos(\alpha)\cdot\cos(\alpha'-\phi^n) \\ &= \cos(\alpha-\alpha'+\phi^n), \end{aligned}\qquad(3)$$

where $\alpha = \alpha(t) = \omega(t)$ is the prediction of solar azimuth based on time (DN1pB neurons), $\alpha' - \phi^n$ is the estimation of the retinotopic solar azimuth (TuBu1 neurons), and $\phi^n$ is the retinotopic direction of a TuBu1 neuron. The above equation suggests that ER4m ring neurons encode the angular difference between the observed (celestial compass) and predicted (clock neurons) solar azimuth, indicating the animal's heading relative to north.

## Complete time compensation

The earth's spin axis is not aligned with the earth's orbit around the sun, causing seasonal shifts in day length and the observed course of the sun across the sky (Fig. 1e). To accurately compute the sun's location at a given time, we need to know the longitudes and latitudes

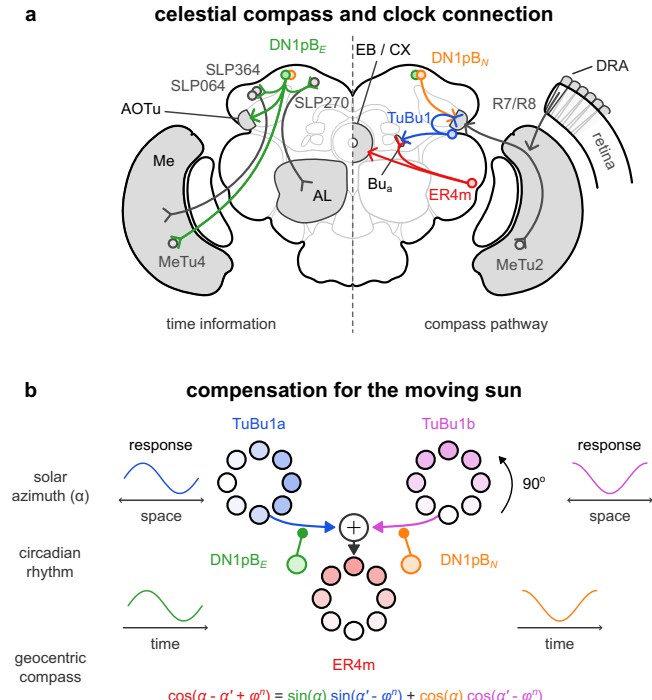

**a | celestial compass and clock connection**

**b | compensation for the moving sun**

$$\cos(\alpha - \alpha' + \varphi^n) = \sin(\alpha)\sin(\alpha' - \varphi^n) + \cos(\alpha)\cos(\alpha' - \varphi^n)$$

**Fig. 3 | A model compensating for the sun's movement in the insect's celestial compass. a** Schematic of the connections between neurons in the brain of *Drosophila melanogaster*. On the left half, we show the first and second-order inputs to the DN1pB (clock) neurons. We show the celestial compass pathway on the right and where the DN1pB neurons join it. Adapted and modified from Gkanias et al. (2023)[34]. **b** The proposed role of the two types of TuBu1 neurons. The spatial encoding of cosine and sine of the solar azimuth across their population is key to the time-compensation mechanism. By combining the activity of the TuBu1 and DN1pB with multiplications and additions, we implement a trigonometric identity and transform the encoding of the solar azimuth into a pattern of activity indicating north. DRA dorsal rim area, Me medulla, AOTu anterior optic tubercle, AL antenna lobe, Bu$_a$ anterior bulb, EB ellipsoid body, CX central complex.

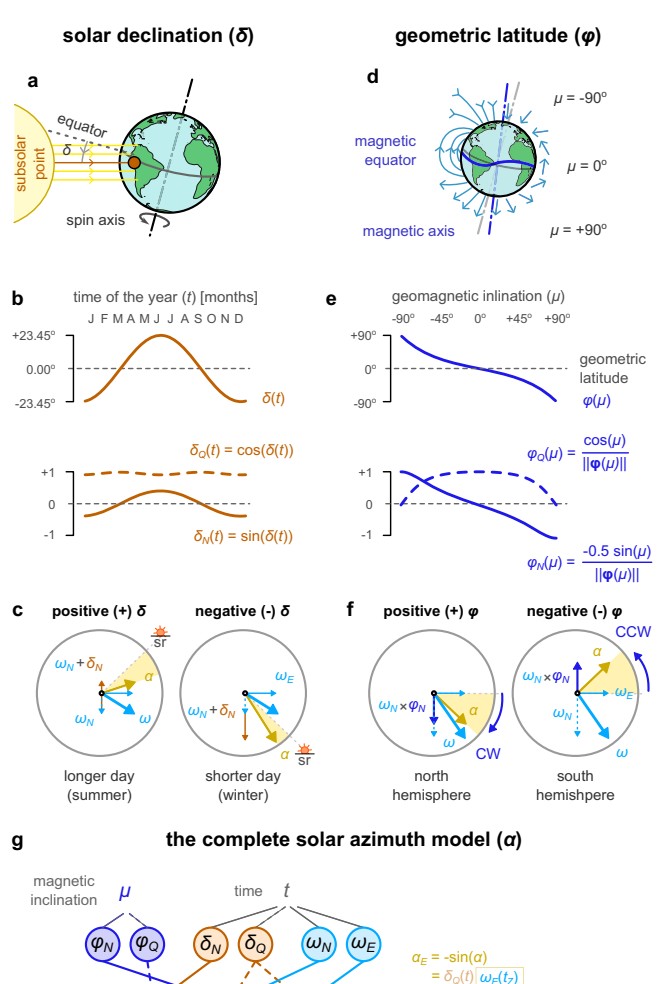

**Fig. 4 | Complete model of the sun's movement. a** The solar declination is the earth's latitude where the sun is exactly at the zenith, which is equivalent to the angle of the line connecting the centres of the earth and sun from earth's equator. **b** The solar declination ($\delta$) is represented by a sinusoidal function of time with an annual period. This can be encoded in a two-dimensional direction as described by its sine ($\delta_N$) and cosine ($\delta_Q$). Vertical axis in arbitrary units. **c** A schematic showing how the solar declination adds to the north-most component of the hour angle, influencing both the sunrise (sr) time and solar azimuth ($\alpha$) based on the season. The horizontal and vertical axes are the sine (east-most) and cosine (north-most component) of the hour angle. **d** The geomagnetic inclination ($\mu$) is the angle between the geomagnetic field and the earth's surface. **e** The geomagnetic inclination is a monotonic function of the geometric latitude ($\varphi$), which can also be described by its sine ($\varphi_N$) and cosine ($\varphi_Q$), representing a two-dimensional direction. Vertical axis in arbitrary units. **f** A schematic showing how the latitude multiplies with the north-most component of the hour-angle. It influences its magnitude and can flip it in the southern hemisphere, allowing for clockwise (CW) and counterclockwise (CCW) movements of the sun. Axes similar to **c**. **g** The full solar azimuth model. The proposed circuit combines information from the geomagnetic inclination and daily and annual clocks, to accurately estimate the sun's course during the day. This estimate is a vector with a north-most ($\alpha_N$) and east-most ($\alpha_E$) component.

of both the subsolar point and the observer[3]. The latitude of the subsolar point is also known as the solar declination ($\delta$; Fig. 4a), which is maximum at 23.45° (the angle of earth's spin axis from a vertical axis) in June, minimum at −23.45° in December, and zero in March and September. This oscillating angle with an annual period can be described by its sine and cosine components (Fig. 4b). We assume that a different pair of clock neurons encode these two components. The first component ($\delta_N$) raises the hour-angle oscillation during summer and lowers it in winter, emulating the longer or shorter days of the year (Fig. 4c) in the northern hemisphere (the relationships would be reversed for the southern hemisphere). Because the solar declination is an angle, we need a second component ($\delta_Q$) to ensure balanced and stable trigonometric computations. Note that solar declination differs from day length ($T_L$), as the latter also depends on the geometric latitude of the observer. Interestingly, knowing any two of the day length, solar declination and observer's latitude, we can estimate the third one (see Supplementary Text S3).

Another way we can estimate the geometric latitude of the observer is by measuring the geomagnetic inclination (or magnetic dip, $\mu$), which is the vertical component of the earth's magnetic field and (approximately) depends on the geometric latitude ($\varphi$) at that point (Fig. 4d)[37]. Monarch butterflies can detect magnetic inclination[28,29], although the existence of this sense in other insects is yet to be established and its pathway to the brain unknown. We assume that another pair of neurons encode the sine ($\phi_N$) and cosine ($\phi_Q$) of the insect's latitude as a function of magnetic inclination (Fig. 4e). $\phi_N$ is positive in the northern

hemisphere and negative in the southern. Thus, we hypothesise that $\phi_N$ is multiplied with the north-most component of hour-angle oscillation ($\omega_N$) to flip it when the insect is in the southern hemisphere and transform its rotation from CW to CCW (Fig. 4f). The role of the other term ($\phi_Q$) is to ensure stable trigonometric computations.

The above information is sufficient to accurately estimate the expected solar azimuth at a specific location and time[3]. A possible

circuit by which these variables could influence the activity of the DN1pB pair so that they now encode the solar azimuth ($\alpha$) rather than the hour angle ($\omega$) is illustrated in Fig. 4g, while the values of $\alpha_N$ and $\alpha_E$ for different combinations of magnetic inclination, solar declination and hour angles are plotted in Supplementary Fig. S3. We refer to this as the 'complete model' of the sun's movement and compare it to the 'hour-angle model' in the following simulations of insect navigation.

## Central-place foraging experiment

To demonstrate the effectiveness of our time-compensated compass, we simulated a foraging experiment for insects such as bees and ants who forage throughout the day to a familiar food site but can take long breaks between foraging trips (spent in their home's darkness). The simulated insects initially perform a random walk searching for food just after sunrise, followed by single return trips to the food location (stored as a vector memory) every hour until sunset. We then tested how an insect using CX navigation would perform in this task with different model compass inputs: (a) without time compensation, (b) using the hour-angle or (c) the complete model. The CX model we used is a modified version of a well-established path integration model[38], which also incorporates vector memories of prominent locations[39]. We then use our proposed compass model as the input heading of the CX with 20% noise. Figure 5a–c shows example foraging routes produced by the three models during a single day.

Overall, no matter which compass model the insect used, it could return home quite accurately during individual foraging excursions (Fig. 5d, green boxes). The sun's movement during a short excursion (a few minutes in this simulation) is not enough to cause noticeable errors in the continuous integration of the home vector. However, longer foraging durations increase the error without time compensation, and also significantly affect the performance with the hour-angle model (as opposed to the complete model), as its error increases linearly with time (Fig. 5e). More significantly, insects without time compensation could not accurately revisit a known food site (Fig. 5a). This is because the food-site location was stored in the memory relative to the sun's position, and as the sun moves, so does this location. Either form of time compensation seems sufficient to navigate back and forth to the food site with relatively good accuracy (Fig. 5b and c). Experiments with desert ants suggest that their homing error is slightly higher than the one our simulations produced using any model[40] (Fig. 5d, green dashed line). Interestingly, the foraging routes of the real animals appear to be significantly more accurate than their homing routes[41], which is only in line with the predictions of the complete model. Despite this advantage of the complete model over the other two, it seems likely that foragers using the hour angle (or even the no-compensation) model combined with other cues, such as visual place recognition in the vicinity of the food site, could produce indistinguishable behaviour (for example, see experiments with honeybees *Apis mellifera* and *A. cerana*[42,43]).

In the above experiments, we used the actual day length for equation (1) (orange lines in Fig. 2c). Supplementary Fig. S4 shows the results when, instead, we use the smoothed day length estimates (black lines in Supplementary Fig. S2), time-shifted to overlay their theoretical values. Although this introduced some extra noise, foraging performance was not dramatically affected.

## Migration experiment

Most foraging insects remain at a similar latitude during their lifetime so it might be expected that the hour-angle model (which adjusts for day length but not solar declination and latitude) would be sufficient for successful behaviour. In contrast, several species of migrating insects travel, within a relatively short time, through multiple latitudes, so it might be expected that they require the complete time-compensation model for their compass system. We therefore simulated insect migration, over realistic travel distances and directions,

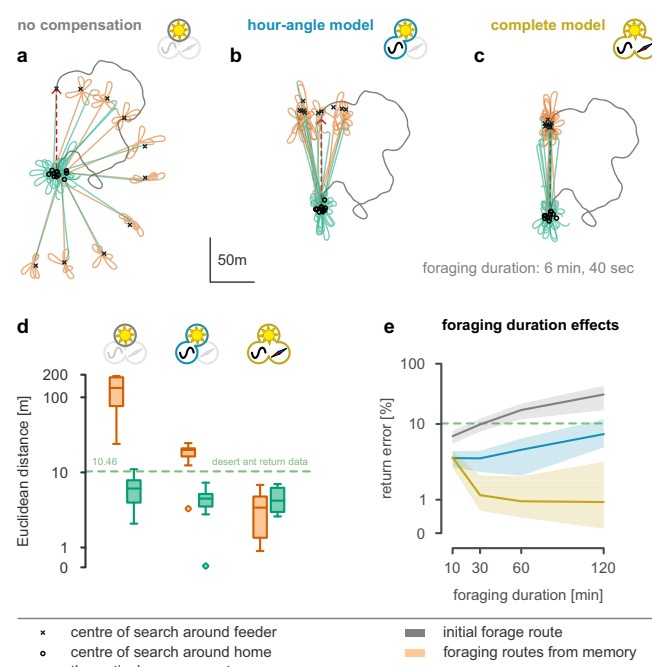

**Fig. 5 | Computer simulated central-place foraging routes.** At dawn, the simulated insect searches for food and stores the food location after finding it. It then returns to its nest. It tries to repeat foraging to the food location every hour until sunset, using (**a**) a model without time compensation, (**b**) the hour-angle model, or (**c**) our complete model. **d**) Euclidean distance (m) of the search centroid from the feeder (red) or the nest (green) using the three models. Each box shows the quartiles of the data (*n* = 9 samples span over Aug 2, 2024; the initial foraging route is excluded). The whiskers extend to show the rest of the distribution except for points that exceed 1.5 × the inter-quartile range, which are marked as outliers. The green dashed line indicates the homing error of desert ants in a similar foraging scenario[40]. **e** Homing error as a function of foraging duration (normalised for foraging distance). Solid lines show the mean error when using the no-compensation (grey), the hour-angle (blue) or the complete model (yellow). Shaded areas are the 95% confidence interval (CI). In all the results there is 20% added compass noise.

using migration routes inspired by three different insect species: the monarch butterfly *D. plexippus*, the Bogong moth *Agrotis infusa*, and globe skimmer dragonfly *Pantal flavescens*. We selected these species to demonstrate migration in different characteristic locations on earth relative to its equator: above (*D. plexippus*), below (*A. infusa*), or across (*P. flavescens*). In this way, we test our model in locations where the sun moves CW, CCW, or switches its moving pattern during the migration.

In the following, we aim to test how travelling long distances across the globe can be done with a time-compensated sun compass, rather than to provide a full account of insect migration. Therefore, in the CX model, we replace its previous path integration or vector memory component with fixed goal coordinates on earth (corresponding to the endpoints of the above migrating species) to test whether the simulated insect can reach them from relevant start points, but use the same speed and daily pattern of movement across all simulations. We modified our simulation to factor in the curvature of the earth's surface. The daily travel capacity was set at eight hours (at 2.5 m sec⁻¹, based on the speed of *D. plexippus*), with one-hour breaks every hour for feeding or rest. We also omitted results without time compensation, as the simulated insects moved in circles around the starting point.

Figure 6a mimics the southward autumn migration routes of monarch butterflies using the hour angle (blue) and complete models (yellow), starting in late August from Mackinac Island (Michigan) and ending in late October in Michoacan (Mexico). Figure 6b shows similar migrations, starting from different locations across the Canadian border and

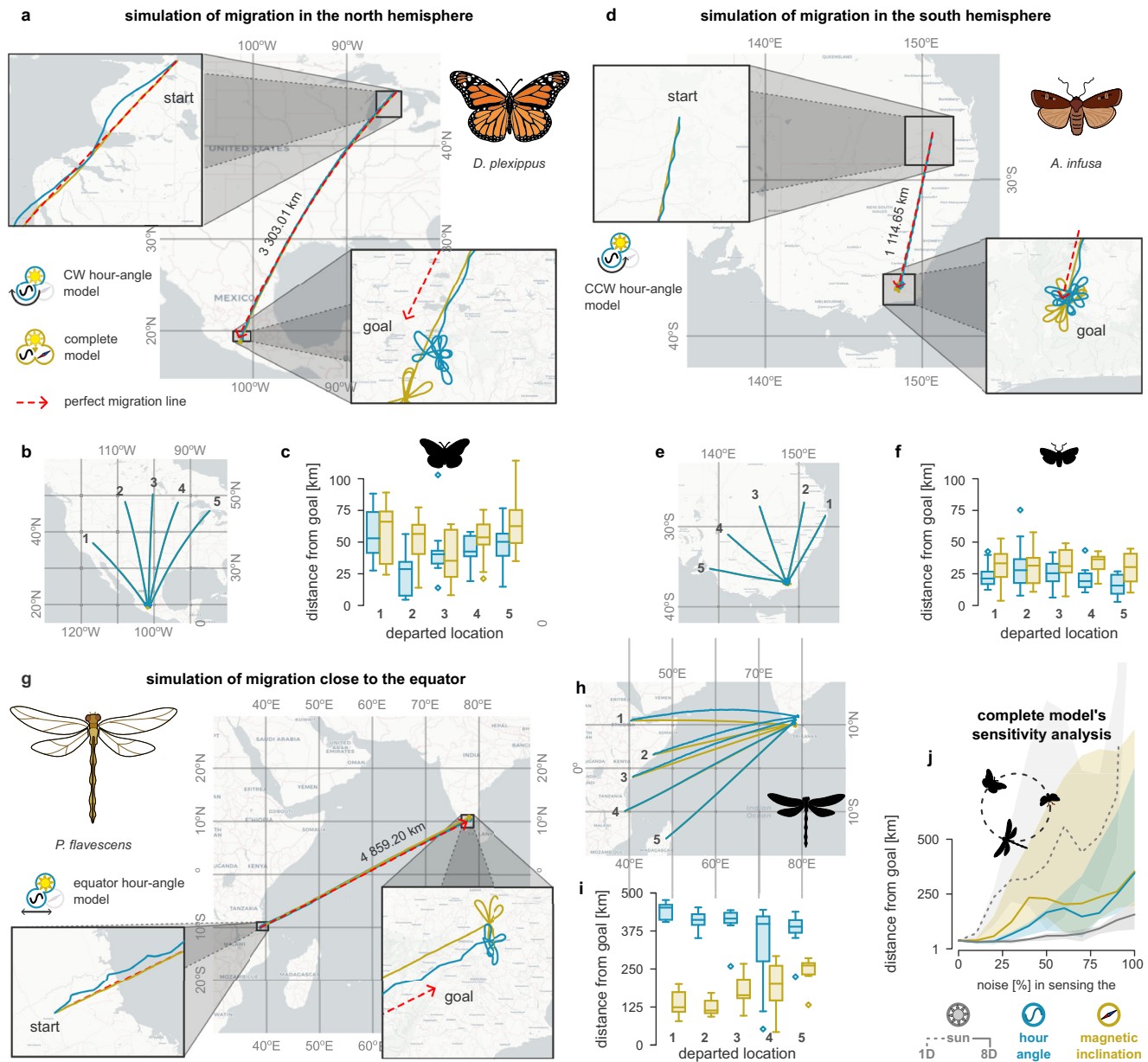

**Fig. 6 | Computer simulations of insect migrations. a** Example simulation of a monarch butterfly (*Danaus plexippus*) during its autumn migration, using a clockwise (CW) hour-angle model (blue) and complete model (yellow). Haversine distance: 3303.01 km. The red dashed arrow illustrates the straight line between the start and goal locations. **b** Examples of similar migrations that start from different locations along the Canadian border and California. **c** Haversine distance (km) between the butterflies' search centroid and the migration's destination point for each model and starting location. Variability comes from spatial (± 5°) and temporal (± 72 h) variations at the beginning of the migration. **d** Similar example simulation for the Bogong moth (*Agrotis infusa*) during its autumn migration, using a counter-clockwise (CCW) hour angle and complete models. Haversine distance: 1114.65 km. **e** Other examples of the same migration, starting from different locations in Australia. **f** Haversine distance between the moths' search centroid and the migration's target location (similar to **c**, but for the moths). **g** Example simulation of a globe skimmer dragonfly (*Pantal flavescens*) during its spring migration,

using the horizontal (east-west) component of the hour-angle model (vertical is set to zero) and the complete model. Haversine distance: 4859.20 km. **h** Similar simulations start from different locations on the east African coast. **i** The Haversine distance (km) of the dragonflies' search centroid from the target migration's target location (similar to **c** and **f**, but for the dragonflies). **j** Sensitivity analysis of the complete model to disturbances in estimating the sun's azimuth (grey; dashed: noise in the one-dimensional angle, solid: noise in the eight-dimensional neural representation of the angle), hour angle (blue) and magnetic inclination (yellow), pooled across the different migrations. In **c**, **f** and **i** each box shows the quartiles of the data (*n* = 10 samples); the whiskers extend to show the rest of the distribution except for points that exceed 1.5 × the inter-quartile range, which are marked as outliers. In **j** the sample size is *n* = 15 per 10% noise and the shaded areas are the 95% confidence interval. In **a-i**, there is 20% added noise in both sensing and processing. All maps were drawn with permission using the Contextly Python package with the CartoDB provider option.

California. Figure 6c summarises the performance of the two models in the above migrations. The variability in the performance comes from variations in the departure location (± 5° in longitude and latitude) and time (± 72 h). Overall, the performance of the two models seems very similar.

Figure 6d shows the simulated routes of Bogong moths during autumn migration from Montrose (Australia) south to Mount Bogong. Figure 6e shows similar migrations starting from different locations in Australia. As this migration occurs in the southern hemisphere, we use the CCW version of the hour angle model. Figure 6f summarises the

migration performance of the two models in the southern hemisphere, showing insignificant differences between the two models. The shorter (compared to butterflies) final distances from the goal are simply because the overall migration distances are shorter so less error accumulates.

Finally, Fig. 6g shows the simulated routes of the globe skimmer dragonflies during their spring migration from Mbekenyara (Tanzania) to Madirai (India), which is the world's longest transoceanic migration (approximately 4859 km). Although dragonflies may fly more continuously during their transoceanic migration, for consistency we used the same migration time pattern as for the monarch butterfly (travel at 2.5 m sec$^{-1}$ with foraging breaks). Exploratory simulations using continuous migrations produced smoother routes but led to the same conclusions. Figure 6h shows similar migrations starting along the east African coast. In these simulations, we use the equator-specific hour angle model. Figure 6i summarises the migration performance using the two models, where we see some more significant differences in their performance. Supplementary Figs. S5, S6 and S7 show the respective migrations of butterflies, moths and dragonflies using alternative hour angle models.

The complete model generally resulted in straighter migration routes than the hour-angle models. However, the deviations induced by the less accurate hour-angle model largely cancel out during the day. As a result, in the single-hemisphere migrations, both models could bring the animals close to their destinations (Fig. 6c, f). The hour-angle model deviates more from the desired destination point for the dragonflies crossing the equator (Fig. 6i), but it can still drive the animal fairly close to its destination. The observed error results from a consistent bias rather than lower precision (Supplementary Fig. S8), which a constant offset could compensate for. Therefore, the only clear advantage of the complete model is that it allows full adaptation to different environments. In contrast, we had to choose the appropriate variant of the hour angle model for simulated migration above, below or across the equator.

We performed further sensitivity analysis by introducing random sensor noise from a uniform distribution, with the upper and lower bounds set as a percentage of the maximum possible measurement value (Fig. 6j). For the hour angle, the error (average distance from goal across all migrating species) becomes significant only for unrealistic levels of noise in estimating the time (more than 25% noise, corresponding to 6 h). Magnetic inclination measurements are expected to be noisier (especially near the equator) but had to exceed 45° to produce noticeable error. If noise in the sun position is introduced independently in each TuBu1 neuron, then its effect averages out across the eight-dimensional encoding of the angle, leading to little change in migration error as the noise level is increased (solid grey line in Fig. 6j). If the noise is added to the sun's angle before TuBu1, the error increases more rapidly, from around 10°) (dashed grey line in Fig. 6j).

As before, these results use the actual day length. Substituting the estimated day length, migrations became slightly more tortuous and less accurate, especially at the start of the migration route (Supplementary Fig. S9). The largest error in the day-length estimates seems to come from the initialisation of $T_L$, which takes some time to converge to the correct value. Although this could be avoided by optimising the initial value as we did with the other parameters, insects might encounter a similar problem.

## Discussion

Insects have a celestial compass that combines the detection of the sun's position and correction for its movement during the day (time-compensation) to obtain a consistent geocentric heading estimate. We suggest that this correction relies on trigonometric principles for spatial and temporal processing, and show a plausible neural mechanism by which the required trigonometry could be implemented. We compare the efficacy of two alternative models for time

compensation: one that makes an approximate correction by assuming a constant change of sun angle with time (the hour-angle model); and one that makes a complete correction for the effects of latitude and declination on the apparent course of the sun relative to an observer in a particular location and time of year. In both cases, the clock is synchronised to solar noon by being reset at sunrise to a value that depends on a running estimate of day length based on skylight irradiance. We test these in central-place foraging and migrating scenarios, showing that the complete model provides only a minor advantage, most noticeably in tasks that require crossing the equator. Both models use information that is in principle available to insects and predict identifiable connectivity and activity patterns in the brain that can be explored anatomically and physiologically.

A striking property of the neural mechanism we propose is that it exploits the fact that both space and time are encoded in the insect brain as sinusoidal activity patterns. For example, solar azimuth information represented by the TuBu1 neurons has a characteristic sinusoidal pattern across a population of neurons[35]. On the other hand, clock neurons (such as DN1pB) are characterised by periodic activity oscillations, usually described as temporal sinusoidal patterns. This characteristic of the activity patterns in the insect brain greatly facilitates trigonometric operations to perform geometric calculations. We note that the same trigonometric operations could be obtained by several alternative circuit configurations that are functionally equivalent to our model. For example, the cosine and sine of the solar azimuth could be encoded in left and right brain hemispheres (Fig. 7a) or implemented by offset connectivity (Fig. 7b). The number of ER5m and TuBu1 neurons could differ (Fig. 7c), as has been observed in the *D. melanogaster* brain[23], and the relative phase of their sinusoids be offset by an arbitrary amount (Fig. 7d).

Our model predicts that the influence of clock neurons on their downstream targets (at least for this circuit) is multiplicative. Traditionally, time information in the insect brain is described in terms of gene expression (*tim*, *per*, *cry*) or the level of proteins mRNA (such as Tim, Per, Cry1, Cry2 and vrille[19]). In a typical clock protein, the mRNA level increases and decreases once daily, but this can vary from tidal to annual periods[20]. With the development of optogenetic tools for *D. melanogaster*, the calcium levels of clock neurons have been observed to follow the same pattern as the proteins, but only daily periods have been described so far. Although calcium is often interpreted as a firing rate signal, it would be useful to clarify how changing calcium levels in clock neurons translates into the membrane potential (for example, following the work on PFNa neurons[44]). This would facilitate understanding how clock neurons affect their downstream targets, which could involve neurotransmitter, neuromodulator or neuropeptide release.

A complete map of clock neurons in fruit flies suggests a circuit architecture where DNs and LNs collect all temporal information and then distribute it to other areas that are responsible for behaviour[19] (Fig. 8). Sensory clocks (from the eyes, ocelli, and antennas) tune neurons in the accessory medulla and other neuropils to synchronise rhythms to external light cues by expressing the Cry1 protein. In our model, we have not mapped the processes before the DNs directly to these anatomical pathways. We suggest, within this upstream circuit of clock neurons (or proteins), we could find processes that (1) calculate the day length, (2) smooth the estimations, (3) calculate the hour-angle and (4) solar declination based on the day and year length, respectively, (5) measure the magnetic inclination, and (6) use it to calculate the insect's latitude. Thus, our model suggests that the different clock neurons are tuned to track 'subtle geophysical forces'[45,46]. More specifically (and testably), we propose that DN1pB neurons combine this information to encode a prediction of the solar azimuth ($\alpha$).

However, our simulations of insect navigation suggest that the time compensation of their celestial compass may not need to incorporate all subtle geophysical forces. The simple assumption of a constant rate of change in azimuth, adjusted for day length, was sufficient for insects migrating large distances within one hemisphere. This

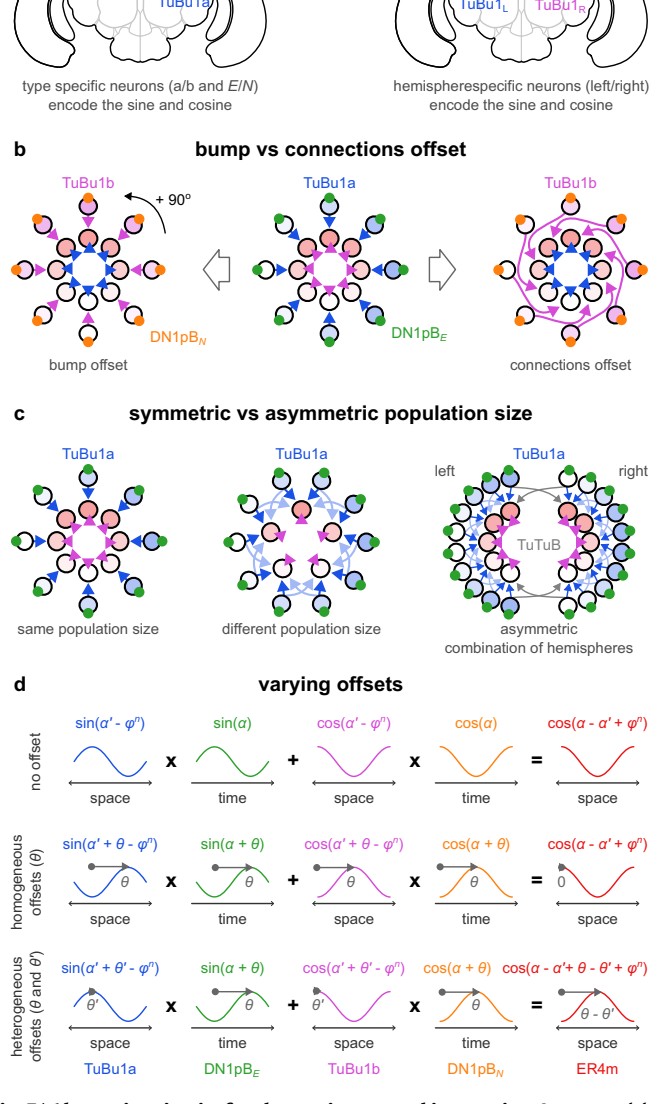

**Fig. 7 | Alternative circuits for the spatiotemporal integration.** In our model, eight pairs of TuBu1a and TuBu1b neurons encode the sine and cosine of the sun's azimuth from the animal's heading (spatial representation) and single $DN1pB_E$ and $DN1pB_N$ neurons encode the sun's azimuth from north (temporal representation). Each pair of TuBu1 neurons (modulated by a DN1pB neuron) targets an ER4m neuron and implements the required trigonometric identity. There are several alternatives to this implementation. **a** A single type of TuBu1 neurons could encode the sine and cosine of the sun's azimuth in the left and right brain hemispheres respectively (adapted and modified from Gkanias et al. (2023)[34]). **b** Two TuBu1 populations could have aligned activity bumps but express the sine and cosine of this angle through a 90° offset in their downstream connectivity. **c** Any relative population size of TuBu1 and ER4m could implement the same trigonometric identity as long as their connectivity is adjusted accordingly. **d** A homogeneous offset of the sinusoids by an arbitrary angle ($\theta$) in TuBu1 and DN1pB does not affect the resulting ER4m sinusoid. Heterogeneous offsets of space ($\theta$) in TuBu1 and time ($\theta' \neq \theta$) in DN1pB produce a constant offset in ER4m, maintaining a consistent geocentric encoding.

linearity assumption diverges most from the actual solar azimuth closer to the equator, since within 23.45° from the equator the sun switches biannually between CW and CCW movement. The complete model, which compensates for both the non-linearity and directional switch, would appear most critical for insects with habitats within this

band, such as globe skimmer dragonflies and tropical forest honeybees *A. mellifera*[47]. However, using an hour-angle model variant for the dragonfly that assumes the sun azimuth switches at noon from directly east to directly west proved almost as effective.

The hour angle model also appears to be sufficiently accurate for central-place foragers. However, there may be some circumstances in which a complete model would be advantageous to reduce the error in return trips to a food source from that shown in Fig. 5b to that shown in Fig. 5c. For example, Saharan desert ants (*Cataglyphis fortis* and *C. bicolor*) that live in featureless salt pans have few other cues available to correct for error and longer foraging durations (inducing more error)[40]. Our results suggested that the foraging capabilities of these insects fall closer to the behaviour produced by the complete model rather than the hour angle (Fig. 5d). Note that in this case, the complete model could be simplified by using constant values for the sine and cosine of latitude, which changes little, rather than requiring latitude to be estimated from magnetic inclination. As such, it is not clear that either foraging or migrating insects need to sense magnetic inclination for time compensation, although this sense may play another role in migration by providing a geographic gradient or cue to arrival at the destination point.

For insects that only need to use their celestial compass in relatively short trips, compensating for the sun's movement only becomes critical when trips to the same location are to be made at different times of the day. In this case, an alternative to using an internal clock for time compensation would be to use familiar visual surroundings, a magnetic compass sense[48,49] or some other constant directional cue to recalibrate the celestial compass at the start of each journey. Experiments with time delays and displacement of ants (*C. fortis* and *C. bicolor*[5]) or honeybees (*A. mellifera*[50,51]) to novel locations make it unlikely that terrestrial visual cues are used. To rule out the possibility that recalibration occurs using a magnetic compass, it is essential to put in conflict their time-compensated celestial compass and their expected magnetic field, which has not yet been attempted.

Implementing a complete time-compensated celestial compass could also be advantageous for robotics, as it provides localisation on earth based on an embedded model of geophysical forces and does not require any satellite-network infrastructure. Even a complete power-down of an autonomous robot, causing loss of time and date information, could be compensated by the proposed methods to estimate time. Apart from the robot's heading, these methods can also provide estimated latitude (calculated as a function of day length and solar declination) and longitude (function of the adaptive hour-angle and a clock tuned to the exact period of the planet's spin). The compass could thus be valuable for outdoor robotics, providing a robust backup when the magnetic field is distorted and the GPS signal is weak; and for planetary exploration[52] by exploiting comparable regularities in the relation of sun position to geographic orientation resulting from orbit and planetary spin.

## Methods
### Optimisation for the day length
To optimise equation (2), we needed the total blue irradiance of the skylight ($I_{sky}$) and the ground-truth day length ($T_L$). Thus, we generated a ground-truth dataset of overall blue skylight irradiance and day length during a calendar year. Then we optimised the free parameters ($\tau_L$, $a$ and $\beta$) of equation (2), using the 'curve_fit' method of the 'SciPy' Python package. The function we optimised was the value of $T_L$ over time, which was calculated using the Euler's method for discrete-time with $dt = 1$ h and $T_L(0) = 7$ h as

$$T_L(t) = \sum_{i=1}^{t} T_L(i-1) + \frac{1}{\tau_L}(a\,I_{sky}(i) + \beta - T_L(i-1)). \quad (4)$$

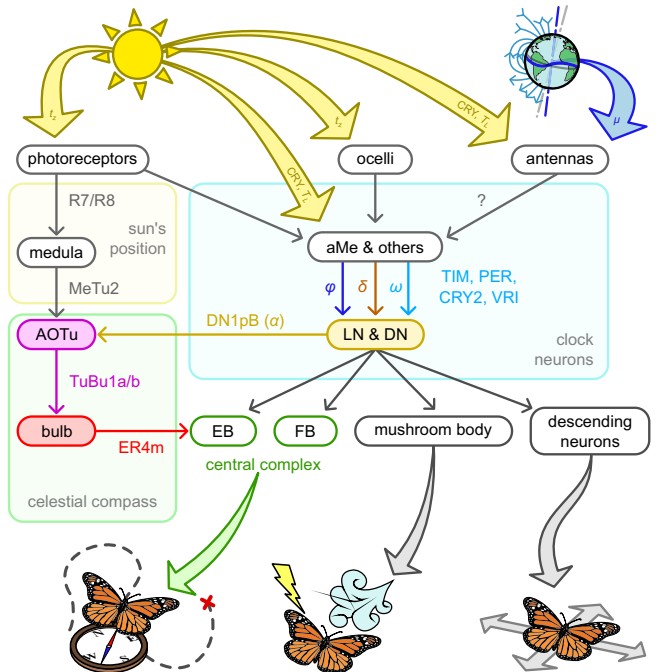

**Fig. 8 | Clock neurons in the insect brain.** Light is detected by the photoreceptors of the compound eyes, ocelli and antennas, and regulates the clock neurons in the insect brain. In our models, these are the zeitgeber time ($t_z$) and day length ($T_L$). The antennae of some insects can also detect the magnetic inclination ($\mu$), which can also play a role in the compass of insects. The celestial compass involves processes through the dorsal medulla (Me), anterior optic tubercle (AOTu), and bulb (Bu), where it forms an activity bump representing the insect's heading relative to geocentric coordinates. The clock neurons in the brain of *Drosophila melanogaster* are the dorsal (DNs) and lateral neurons (LNs), which receive indirect input from the photoreceptors, ocelli and antennas, through the accessory medulla (aMe) and other areas. We suggest that their inputs include the insects' hour angle ($\omega$), latitude ($\phi$), and the solar declination ($\delta$). DNs and LNs target different major areas in the insect brain that control the insect's behaviour. These include the ellipsoid body (EB) and fan-shaped body (FB) of the central complex (CX), the mushroom bodies, and the descending neurons. A particular type of DN, the DN1pB, targets the AOTu and the TuBu1 neurons and therefore is part of the celestial compass pathway. We suggest that this neuron encodes a prediction of the solar azimuth (relative to an absolute coordinate), which integrates with the detected egocentric solar azimuth into a geocentric compass. Shaded nodes in the architecture denote areas modelled in this work. Grey arrows and empty boxes were not directly modelled in this work.

We calculate day length and overall irradiance using the 'skylight' Python package and the sky model described by Vévoda et al. (2022)[53]. The observer was set to be in Edinburgh (55.9533°N, 3.1883°W), from January 1 to December 31, 2024, collecting one sample per hour (8760 samples in total). For each sample, the day length ($T_L^*$) was calculated as the time between sunrise and sunset (in hours) of the respective day. For the overall skylight intensity ($I_{sky}$), we estimated the irradiance by using 1000 homogeneously distributed rays from the simulated sky, we extracted the visible-blue light irradiance and calculated the average across rays.

To simulate insect foraging patterns, we randomly selected a $T_F/12$ h proportion of the 8760 homogeneous $I_{sky}$ samples of the year, where $T_F$ is the foraging time. The remaining samples were set as $I_{sky} = 0$ Wm$^{-2}$sr$^{-1}$, simulating the time spent inside. The above method produces random light exposures daily, with on-average $T_F$ per day within a year. This means we might have 12 h of foraging spread over a day and no foraging on another day. For foraging experiments, $T_F \in [1$ h, 2 h, 4 h, 8 h], while for migration, $T_F = 12$ h.

## The celestial compass model

Our results assume a compass model that accurately returns the solar azimuth. We assume that this is represented in the AOTu by 16 TuBu1 neurons with preferred directions centred at homogeneously distributed angles in a ring,

$$\phi^n = n\,22.5^\circ, \qquad \text{where} \quad n \in \{1, \dots, 16\}. \tag{5}$$

These take input from the medulla MeTu2 neurons, whose responses are calculated as

$$\text{MeTu2a}^n(t) = \sin(\alpha_{sun}(t) - \theta(t) - \phi^n) + \epsilon_{sun}^n(t), \tag{6}$$

$$\text{MeTu2b}^n(t) = \cos(\alpha_{sun}(t) - \theta(t) - \phi^n) + \epsilon_{sun}^n(t), \tag{7}$$

where $\theta(t)$ is the heading of the animal at time $t$ (hours), $\phi^n$ is the preference angle of the $n^{th}$ MeTu2 neuron, $\alpha_{sun}(t)$ is the solar azimuth, and $\epsilon_{sun}^n(t) \sim \mathbb{U}(-\eta, \eta)$ is uniform noise in the encoding of the sun's position, which is proportional to the given noise factor ($\eta \in [0, 1]$).

For the one-dimensional noise in our sensitivity analysis, we instead included the noise in the angle itself as

$$\text{MeTu2a}^n(t) = \sin(\alpha_{sun}(t) - \theta(t) - \phi^n + \epsilon_{sun}(t)), \tag{8}$$

$$\text{MeTu2b}^n(t) = \cos(\alpha_{sun}(t) - \theta(t) - \phi^n + \epsilon_{sun}(t)). \tag{9}$$

Note that now the same noise applies to all the neurons.

## Models of compensation for the moving sun

Three models of the compass are used in different parts of our results: the no-compensation, hour-angle, and complete models (in order of ascending complexity). Each model produces the responses of the ring neurons (ERs), which are input to the CX model.

**The no-compensation model.** The activity of the ERs is equivalent to the output of the optic lobe's medulla-tubercle (MeTu) neurons,

$$\text{ER4m}^n(t) = \text{TuBu1b}^n(t) = \text{MeTu2b}^n(t). \tag{10}$$

**The hour-angle model.** Here, we introduce a basic correction using the hour-angle. The responses of the DN1pB clock neurons relate to the hour angle as

$$\text{DN1pB}_E(t) = \omega_E(t), \tag{11}$$

$$\text{DN1pB}_N(t) = \omega_N(t), \tag{12}$$

where $t$ is the time (hours) since the beginning of the year. The components $\omega_E$ and $\omega_N$ are calculated as

$$\omega_E(t) = -\sin(\omega(t)) + \epsilon_{time}(t), \tag{13}$$

$$\omega_N(t) = -\cos(\omega(t)) + \epsilon_{time}(t), \tag{14}$$

where $\omega(t)$ is computed by equation (1), and $\epsilon_{time}(t) \sim \mathbb{U}(-\eta, \eta)$ represents the uniform noise in the encoding of time, which is proportional to the given noise factor ($\eta \in [0, 1]$). Note here that we assume a perfect estimate of the sunrise time ($t_{sr}$; used in equation (1) to estimate the zeitgeber time). In migrations in the southern hemisphere, we use the CCW hour angle model, where we set $\omega_N(t) = \cos(\omega(t)) + \epsilon_{time}(t)$. For the migrations across the equator, we set $\omega_N(t) = \epsilon_{time}(t)$.

The activity of the tubercle-bulb (TuBu) neurons is then estimated as

$$TuBu1a^n(t) = -DN1pB_E(t) \cdot MeTu2a^n(t), \qquad (15)$$

$$TuBu1b^n(t) = -DN1pB_N(t) \cdot MeTu2b^n(t). \qquad (16)$$

TuBu neurons add up column-wise in the bulb, where they target the ERs, whose activity is calculated as

$$ER4m^n(t) = TuBu1a^n(t) + TuBu1b^n(t). \qquad (17)$$

**The complete model.** The complete model also involves solar declination and the insect's latitude. Solar declination is computed as

$$\delta(t) = 23.45° \sin\left(\frac{284 + t/24}{365} \, 360°\right), \qquad (18)$$

where $t$ is the time (hours) since the start of the year, and it is encoded by the north-most and equator-most components as

$$\delta_N(t) = \sin(\delta(t)), \qquad (19)$$

$$\delta_Q(t) = \cos(\delta(t)). \qquad (20)$$

Geometric latitude can be approximated as a function of geomagnetic inclination[37] as

$$\phi(\mu, t) = -\tan^{-1}(0.5 \tan(\mu(t) + \epsilon_\mu(t))), \qquad (21)$$

where $\mu(t)$ is the local geomagnetic inclination, and $\epsilon_\mu(t) \sim \mathbb{U}(-\eta\,90°, \eta\,90°)$ is uniform noise in sensing the magnetic inclination, based on the noise parameter $\eta \in [0, 1]$. The north-most and equator-most components encode the latitude as

$$\phi_N(\mu, t) = \sin(\phi(\mu(t))), \qquad (22)$$

$$\phi_Q(\mu, t) = \cos(\phi(\mu(t))). \qquad (23)$$

Using the above information, the apparent solar azimuth can be estimated as[3]

$$\alpha_E(\mu, t) = \delta_Q(t)\,\omega_E(t), \qquad (24)$$

$$\alpha_N(\mu, t) = \phi_Q(\mu, t)\,\delta_N(t) + \phi_N(\mu, t)\,\delta_Q(t)\,\omega_N(t), \qquad (25)$$

corresponding to the solar azimuth's east (negative sine) and north (negative cosine) components.

We replace the responses of the DN1pB clock neurons with the solar azimuth components,

$$DN1pB_E(t, \mu) = \alpha_E(\mu, t), \qquad (26)$$

$$DN1pB_N(t, \mu) = \alpha_N(\mu, t). \qquad (27)$$

We simulate the observed magnetic inclination ($\mu$) using the actual geometric latitude of the animal as

$$\mu(t) = \tan^{-1}\left(\frac{-2\sin(\phi(t))}{\cos(\phi(t))}\right). \qquad (28)$$

The remaining calculations are the same as in the hour-angle model.

## The central complex model

As we do not focus on the dynamics of the ring attractor or the specific neural response of the neurons in CX, we simplified the model described in Stone et al. (2017)[38] by replacing the processing layers with vectors represented by complex numbers. Thus, we transformed the compass representation of the ER4m neurons as

$$z_{ER4m}(t) = \frac{1}{16}\sum_{n=1}^{16} r_{\alpha'}^n(t)\,e^{i\phi^n} + \epsilon, \qquad (29)$$

where $\epsilon = \epsilon_x + i\epsilon_y \sim \mathbb{U}(-\eta, \eta) \in \mathbb{C}$ is a random number drawn from a uniform distribution, with real and imaginary components in the range $[-\eta, \eta]$, and $\eta \in [0, 1]$ is the selected noise level. By default, in all our experiments $\eta = 0.2$.

The representation of ellipsoid-body protocerebral-bridge gall (EPG) neurons can then be approximated as

$$z_{EPG}(t) = \frac{z_{ER4m}(t) + \epsilon}{|z_{ER4m}(t) + \epsilon|}. \qquad (30)$$

Note that we need to normalise this complex number to ensure that we only keep the direction information, which is the estimated heading of the animal.

Multiplying the heading with the speed of the animal ($v$), we compute its velocity in the protocerebral-bridge FB nodulus (PFN) neurons,

$$z_{PFN}(t) = z_{EPG}(t)\,v(t). \qquad (31)$$

We use the activity pattern of the PFNs to update the working memory (M) of the CX as

$$\tau_M \frac{dz_M}{dt} = z_{PFN}(t), \qquad (32)$$

where $\tau_M = 40$ km is the time-constant of the memory charge. Similarly, we have a target memory ($z_G \in \mathbb{C}$) that can be used as the migration target or the foraging site in our experiments, and by default it is zero.

The allocentric goal direction is computed by FB columnar (FC) neurons, which we implement as

$$z_{FC2}(t) = \frac{z_G(t) - z_M(t)}{|z_G(t) - z_M(t)|}. \qquad (33)$$

Note that here we also normalise with the magnitude of the complex number, as this population of neurons represents the allocentric direction only and not the distance of the goal location.

Finally, the egocentric steering signal is decomposed into two axes, at 45° towards the left (L) or right (R), and it is calculated as

$$z_{PFL3,L}(t) = z_{FC2}(t) - z_{EPG}(t)\cos(-45°)\,e^{-i45°} + \epsilon, \qquad (34)$$

$$z_{PFL3,R}(t) = z_{FC2}(t) - z_{EPG}(t)\cos(45°)\,e^{i45°} + \epsilon. \qquad (35)$$

## Simulations

We run a set of simulations for migrating and central-place foraging insects. The update of the heading direction and position of the animals in all the simulations occurs in the same way.

Based on the above CX model, we calculate the angular velocity of the animal as

$$\frac{d\theta}{dt} = \frac{1}{4}\left(|z_{PFL3,L}(t)| - |z_{PFL3,R}(t)|\right), \tag{36}$$

where $\theta(t)$ is the heading of the animal. Then we update the linear velocity of the animal as

$$\frac{dz_{xy}}{dt} = v(t)\,e^{i\theta(t)}, \tag{37}$$

which is used for updating its actual position, $z_{xy}(t)$. The speed in the above equation ($v$) corresponds to the approximate speeds of the insects in the following experimental scenarios.

**Central-place foraging.** For these experiments, we placed the insects in Edinburgh (55.9533°N, 3.1883°W), the United Kingdom, on August 2, 2024. We use local coordinates where the nest is at point zero, which is also the initial location of the insect. The insect's speed was constant at $v(t) = 0.5\,\text{m sec}^{-1}$ for all $t$ the insect was moving, or $v(t) = 0\,\text{m sec}^{-1}$, for those in which the insect was resting. This experiment had three phases: searching for a food source, homing, and foraging to a known food site.

A random route is generated in the initial phase, and the animal is forced to follow it (see 'Defining a random foraging route'). The CX model is updated in each step using the current direction and speed, as computed by the difference between two subsequent points on the route. The final location of the route is stored as the goal vector memory. The insect then alternates the homing and foraging phases, with an hour of rest after every homing phase.

In the homing phase, the goal location is set to the zero point (home), and we update the angular and linear velocity of the animal using equations (36) and (37). When the insect nears its goal, the CX model produces a characteristic search pattern. In each step, we estimate the probability of the animal expressing such a pattern by detecting four consecutive turning points (see 'Detecting a turning point'). The centroid of the four turning points approximates the centre of the search, which we interpret as the expressed goal location of the insect

$$z_{\text{centroid}} = \frac{1}{4}\sum_{c=1}^{4} z_{\text{turn}}^c. \tag{38}$$

The foraging phase is similar to the homing, but we replace the goal location with the stored vector memory of the known food site.

**Migration.** For the migration experiments, we assume that insects can travel for a maximum of 8 h per day (only between sunrise and sunset), they need to stop every hour for rest and feeding, and that their stops last for 1 h. The speed of the insects was set to $v(t) = 2.5\,\text{m sec}^{-1}$, which is based on the speed of *D. plexippus* monarch butterflies (although this could differ in reality between insects), and the time step used was d$t = 50$ min.

The simulation of autumn migration (from August 29 to October 31, 2024) of monarch butterflies (*D. plexippus*) started close to Mackinac Island (Table 1, departure point 5), Michigan, with a destination close to Michoacan, Mexico, which is 3303.01 km long. The spring migration (from March 29 to May 31 2024) was in the opposite direction. Winter (from November 21, 2024, to January 23, 2025) and summer migrations (from June 21 to August 23, 2024) were from the same locations as autumn and spring, respectively. The autumn migration (from September 4 to October 1, 2024) of Bogong moths (*A. infusa*) started close to Montrose (Table 1, departure point 2), Australia, with a

**Table 1 | Migration departures and destinations**

| Animal (season) | Location | Depart from | | Destination | |
|---|---|---|---|---|---|
| Butterfly (autumn) | 1 | 36.91°N | 116.76°W | 19.55°N | 101.60°W |
| | 2 | 48.24°N | 107.95°W | 19.55°N | 101.60°W |
| | 3 | 50.32°N | 100.31°W | 19.55°N | 101.60°W |
| | 4 | 48.06°N | 93.41°W | 19.55°N | 101.60°W |
| | 5 | 45.76°N° | 84.72°W | 19.55°N | 101.60°W |
| Moth (autumn) | 1 | 28.64°S | 153.29°E | 36.84°S | 148.46°E |
| | 2 | 27.00°S | 150.65°E | 36.84°S | 148.46°E |
| | 3 | 27.48°S | 145.10°E | 36.84°S | 148.46°E |
| | 4 | 31.00°S | 141.16°E | 36.84°S | 148.46°E |
| | 5 | 35.24°S | 138.91°E | 36.84°S | 148.46°E |
| Dragonfly (spring) | 1 | 11.01°N | 40.38°E | 10.00°N | 78.00°E |
| | 2 | 3.12°N | 45.61°E | 10.00°N | 78.00°E |
| | 3 | 2.04°S | 40.84°E | 10.00°N | 78.00°E |
| | 4 | 10.00°S | 39.00°E | 10.00°N | 78.00°E |
| | 5 | 16.28°S | 48.54°E | 10.00°N | 78.00°E |

destination close to Mount Bogong, which is 1114.65 km long. The spring migration (from March 4 to April 1, 2024) was in the opposite direction. Winter (from November 27 to December 24, 2024) and summer migrations (from May 27 to June 24, 2024) were from the same locations as spring and autumn, respectively. The spring migration (from February 4 to May 1, 2024) of globe skimmer dragonflies (P. flavescens) started close to Mbekenyera (Table 1, departure point 4), Tanzania, with a destination close to Madurai, India, which is 4859.20 km long. The autumn migration (from September 4 to December 1, 2024) was in the opposite direction. Winter (from November 27, 2024, to February 23, 2025) and summer migrations (from April 29 to July 24, 2024) were from the same locations as autumn and spring, respectively. Other departure points for all the insects are listed in Table 1. Before each simulation, we add $\pm 5°$ noise at the departure point and $\pm 72$ h noise at the departure time.

For the foraging experiments, we treat the world as a two-dimensional plane. For migration, it is necessary to consider the earth's curvature; hence the equations for location and motion must be modified. The initial heading of the animal was calculated as

$$\theta(t_0) = \tan^{-1}\left(\frac{\sin(\Delta\lambda)\,\cos(\phi_e)}{\cos(\phi_s)\,\sin(\phi_e) - \sin(\phi_s)\,\cos(\phi_e)\cos(\Delta\lambda)}\right),$$

and the distance between two points on earth (haversine distance) was calculated as

$$\rho = 2R\tan^{-1}\left(\frac{\sqrt{\sin^2(\frac{\Delta\phi}{2}) + \cos(\phi_s)\,\cos(\phi_e)\,\sin^2(\frac{\Delta\lambda}{2})}}{\sqrt{1 - \sin^2(\frac{\Delta\phi}{2}) + \cos(\phi_s)\,\cos(\phi_e)\sin^2(\frac{\Delta\lambda}{2})}}\right)$$

where $R = 6\,378\,137$ m is the radius of earth and

$$\begin{aligned}
\lambda_s &= \text{start longitude}, & \phi_s &= \text{start latitude}, \\
\lambda_e &= \text{target longitude}, & \phi_e &= \text{target latitude}, \\
\Delta\lambda &= \lambda_e - \lambda_s, & \Delta\phi &= \phi_e - \phi_s.
\end{aligned} \tag{39}$$

where west and south directions were represented as negative angles.

The location of the animal on earth was transformed into a complex number for consistency as

$$z_{xy}(t) = \phi(t) + i\,\lambda(t). \tag{40}$$

Thus, in the CX model, the goal location for the migrating experiments is set as

$$G(t) = \rho\, e^{i\theta(t_0)}. \tag{41}$$

We transformed the location and direction of the animal into a quaternion, $q_{xy\theta}$, to facilitate spherical computations. We used the 'Rotation' package of the SciPy library in Python to do this (see the code for details). So steering was applied as

$$q_{xy\theta}(t) = q_{xy\theta}(t - dt)\left(\cos\left(\frac{1}{2}\frac{d\theta}{dt}\right) - k\,\sin\left(\frac{1}{2}\frac{d\theta}{dt}\right)\right), \tag{42}$$

and forward movement as

$$q_{xy\theta}(t) = q_{xy\theta}(t - dt)\left(\cos\left(\frac{v(t)\,dt}{2R}\right) + j\,\sin\left(\frac{v(t)\,dt}{2R}\right)\right), \tag{43}$$

where j and k are two of the imaginary parts in the quaternion. We always update the heading before moving forward, which composes one step in the simulation. Using the same SciPy package, we can transform the quaternion back to the coordinates of the animal on earth and its heading direction. The coordinates are then transformed into a complex number using equation (40).

Each simulation is run from the start to the end date of the migration, and the insects are allowed to travel only between sunrise and sunset.

**Defining a random foraging route.** The initial search for food was created using a von Mises distribution and Newtonian physics. The starting point was set as the home location, $z_s = 0$, and the final point (food source) was 100 m towards the east, $z_e = 100$. We drew 25 000 $\left(\frac{5 \cdot 100\,\text{m}}{0.5\,\text{m}\,\text{sec}^{-1} \cdot 1\,\text{km}}\right)$ bearing directions for the path from von Mises distribution as

$$\frac{d\theta}{dt} \sim \text{VonMises}(\mu = 0, \kappa = 100), \tag{44}$$

and low-pass filtered to smooth the turns. Subsequently, the position was updated using equation (37). The generated positions were resized and rotated to end at the final point as

$$z_{xy}(t) = \frac{z_{xy}(t)}{z_{xy}(t_e) - z_{xy}(t_s)}\,(z_e - z_s). \tag{45}$$

The positions were resampled every 0.5 m sec$^{-1}$ using linear interpolation. The final heading directions were then calculated as $\angle\frac{dz_{xy}}{dt}$, $\forall\ t > t_0$, where $dt = 1$ km.

**Detecting a turning point.** To detect the search pattern of an insect, we first needed to detect whether the insect changed its heading direction sufficiently. We mark a sufficient change in the heading when its difference from 25 m before is more than 120° and no other turning point was detected in the past 50 m.

**Definition of the desert-ant return data**
In Huber et al. (2015)[40], they measured the distance between the desert-ant search area and their fictive nest ($y$), as a function of their foraging distance ($x$), and fit a quadratic equation to allow interpolation for any distances,

$$y = 2 \cdot 10^{-5}x^2 + 0.083x + 1.96\,\text{m}. \tag{46}$$

In our analysis, we use this equation to compare our simulated results to the ones produced by animals.

**Solar ephemeris.** The sun's course during the day depends on the location of the animal on earth and the time of the year. To calculate these, we use the 'skylight' Python package, which implements the solar ephemeris suggested by the Global Monitoring Laboratory of the US National Oceanic and Atmospheric Administration.

**Performance evaluation**
Given the total distance travelled ($C$) and the straight-line distance of the insect from the goal location ($L$), the tortuosity of the path at a specific time ($t$) is

$$\varsigma(t) = \frac{C(t)}{L(t)}. \tag{47}$$

The Euclidean distance of the insect from its goal location is calculated as

$$\varepsilon_z = |z_{xy} - z_{\text{goal}}|. \tag{48}$$

In Figs. 6 and 5, all measurements were taken when the insect was closest to its centre of search.

Similarly, the error between the estimated ($T_L$) and actual day length ($T_L^*$) is calculated as

$$\varepsilon_{T_L}(t) = |T_L(t) - T_L^*(t)|. \tag{49}$$

**Software**
We used custom code for our simulations and data analysis. This was written in Python 3.9.7 using the PyCharm v2024.1 (Professional Edition) IDE, and open-source packages including: NumPy (v1.26.3), SciPy (v1.12.0), Skylight (v1.0b2), Pandas (v2.1.4), OpenPyXL (v3.1.5), Matplotlib (v3.8.0), Seaborn (v0.12.2), Contextly (v1.5.1) and Loguru (v0.5.3). The maps were created using the Contextly (v1.5.1) Python package with the CartoDB provider option. All clip art and data in the figures were created or reformatted using the Inkscape (v1.3.2) software.

**Reporting summary**
Further information on research design is available in the Nature Portfolio Reporting Summary linked to this article.

## Data availability
Source data are provided with this paper.

## Code availability
The code that runs all the simulations and generates all the plots is publicly available through Code Ocean (https://doi.org/10.24433/CO.7664921.v1).

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

## Acknowledgements

This work was financially supported by the European Union, Horizon Europe (Project 101046790, InsectNeuroNano), and Horizon Europe Guarantee (UKRI grant 10032249). Thanks to Stanley Heinze for the insightful discussion on the insects' clock inputs, to James Foster for the discussion on honeybee time compensation, and to Robert Mitchell for his helpful comments on the manuscript.

## Author contributions

Evripidis Gkanias: conceptualisation, formal analysis, investigation, methodology, project administration, software, validation, visualisation, writing—original draft, writing—review and editing. Barbara Webb: conceptualisation, funding acquisition, investigation, methodology, project administration, resources, supervision, validation, writing—review and editing.

## Competing interests

The authors declare no competing interests.
