## [Transparent Peer Review file · Nature Communications]

Spatiotemporal computations in the insect celestial compass

Corresponding Author: Dr Evripidis Gkanias

Version 0:

Reviewer comments:

Reviewer #1

(Remarks to the Author)

This paper addresses an obvious problem in animal navigation using a celestial compass: time compensation. Since celestial objects move according to the time of the day and the season, without time information the direction encoding can be very wrong. In addition, the viewing perspective of the celestial objects also depends on the latitude of the globe.

The paper starts by constructing a model that represents time based on the CRY1 protein expression found in the monarch butterflies. In this model, the insect estimates the daylength from the skylight irradiance through the CRY1 protein level. Then based on the daylength, it's straightforward to interpret the "sun dial" information. To encode the time information from this daylength model, the authors hypothesize that two clock neurons found in fruit flies represent modulation of the expected solar azimuth collectively. Specifically, these DN1pB corresponds to two orthogonal directions of the solar azimuth. Based on previous studies, they further speculate that two types of TuBu1 neurons store the expected solar azimuth in the form of sin and cos (90deg phase offset). Finally, the solar declination and geometric latitude were modelled separately to account for season and location. Putting all these together, the authors produced a model that can track an absolute direction based on solar azimuth independent of time and place. This model was put to test in two simulation experiments to demonstrate the accuracy with simple daytime compensation and with complete model compensation. If the authors feel that the above summary is incorrect somewhere, it would be good to clarify that somewhere, as this is my understanding from reading the paper.

In general, this paper provides a comprehensive theoretical derivation on how time-compensated celestial navigation could work in insects. However, due to some knowledge gaps from the current literature, several mechanistic speculations had to be made in both the exact roles of certain neurons and the way solar azimuth is represented in the neurons. As a reader, I would like to see more demonstration on the "alternative hypotheses" of these speculations. For instance, encoding a variable with orthogonal dimensions is convenient computationally but may not be how biological systems work. For TuBu1a and TuBu1b, is the 90deg offset necessary for the circuit to work or can they be in any known offset? Is there an alternative configuration for Fig 1h?

The central-place foraging validation is good. I understand that the agent performs a random walk initially in each trial, so the navigation circuit is challenged by a different homing initial condition (and directional integration). It would be useful to mark the accuracy as measured experimentally in some foraging insects on the figure for reference. For the migration experiment, it seems like the agent starts from the same location every time, so the only variation is the 20% added compass noise. This is likely not a good enough representation of natural variation. Perhaps the authors should consider initializing the migration with at different time of the day, season, or minor changes in location. It would be interesting to know if the navigation model performs better at any time or location. If the sole objective was to show that equator crossing requires complete compensation, does it require a simulation as the sign flip is so drastic in the model? My suggestion would be to explore how to add value to the migration simulations. For instance, is the direction variation in the simulation consistent with experimentally measured migration paths, or direction choices in navigating insects. It would be informative to build a simple phenomenological model of navigation based on experimental data and compare to this mechanistic model of navigation. It would be also informative to see under what other conditions the model might fail (if any).

Overall, this is a very nice paper and would be informative to the community.

(Remarks on code availability)

Reviewer #2

(Remarks to the Author)

This manuscript describes a speculative model by which insects navigating via celestial cues might compensate for the movement of the sun during the day. The authors have built a navigation model that combines input from solar position, timekeeping neurons, and magnetic fields. They develop a simple model based on the hour angle of the sun, and more complete model that adds an estimate of latitude acquired via geomagnetic inclination. They then simulate place foraging over periods of hours, and distant migrations that last over months, to demonstrate approximately successful navigation.

To their credit, the authors have worked to keep the model based on input and circuits that real insects are known to use, such as time clock neurons and magnetic inclination. Every step seems plausible, at least in some insects, and aimed at explaining behavior that is observed, but not fully understood. To this end the models do offer believable mechanisms by which solar time compensation could be achieved in insects.

My main concern is that this process requires many steps of neural calculations that aren't directly tested. I have little sense of how likely this particular model is, and for which insects. Further, estimates of magnetic inclination, necessary for the full model, seem particularly tricky to me. The only citation given to support this is for the monarch butterfly, which curiously migrated fine in their simulations with just the hour-angle model. Does the globe skimmer dragonfly possess this sense? The authors seem to claim that it must, as it crosses the equator, which required the full model. But magnetic inclination is especially low around the equator--how well could an insect estimate this variable, and how sensitive would it be to local or secular magnetic variation? Other than the comparison of complete and hour-angle models, the model lacks a rigorous sensitivity analysis. Many inputs, such as sun position, time-estimate of sunrise, and magnetic field inclination, are certainly subject to error (such as by clouds). How much does this matter? My concern is that a complex model with many free parameters and little ground truthing might need much revision as our specific knowledge improves.

Minor:

280: "This describes an oscillation with an annual period, which can be further decomposed into two sinusoidal functions (Fig. 2b)."

-This isn't what I normally think of as "decomposing" a sinusoid, since the cosine of this sinusoid has double its frequency. In other words it is not in terms of simpler functions. But further, the motivation for this step isn't clear to me, other than that it yields functions you want to use later?

Fig 2

-Many axes are not labeled on the figure, or adequately described in the legend or main text. For example, in 2c it is quite difficult to determine what is plotted when units are missing from both horizontal and vertical axes.

430 "Finally, Figure 4e-g shows the simulated routes of the globe skimmer dragonflies during their spring migration from Mbekenyara (Tanzania) to Madirai (India), which is the world's longest transoceanic migration (approximately 4859 km)."

-I don't understand, did this simulated migration also include "daily travel capacity of insects is eight hours (at 2.5 m sec⁻¹, based on the speed of *D. plexippus*), and they need one-hour breaks every hour for feeding or rest", even though those aren't part of this migration?

(Remarks on code availability)

Version 1:

Reviewer comments:

Reviewer #1

(Remarks to the Author)

I am happy with the revision in this version as it addresses most of my concerns. The authors have put in great work in refining the analysis.

(Remarks on code availability)

Reviewer #2

(Remarks to the Author)

The authors have done an excellent job of addressing my comments, and I'm especially pleased with the addition of a sensitivity analysis. My only remaining point of confusion is with my final comment, where the authors seemed to simulate 8 hour daily travel and hourly rest periods for the globe skimmer migration between East Africa and India. They now comment that continuous travel makes little difference to the results of the model, and skipping the rest period produced no significant changes. But these details are just inserted without specific reference to the globe skimmers. The reason for my initial confusion was that I was unsure if these parameters were the same for the globe skimmer migration, so I think it would be most helpful to mention that specifically when discussing the dragonfly.

(Remarks on code availability)

Manuscript NCOMMS-24-42339

Response to reviewers

Reviewer #1 (Remarks to the Author):

This paper addresses an obvious problem in animal navigation using a celestial compass: time compensation. Since celestial objects move according to the time of the day and the season, without time information the direction encoding can be very wrong. In addition, the viewing perspective of the celestial objects also depends on the latitude of the globe.

The paper starts by constructing a model that represents time based on the CRY1 protein expression found in the monarch butterflies. In this model, the insect estimates the daylength from the skylight irradiance through the CRY1 protein level. Then based on the daylength, it's straightforward to interpret the "sun dial" information. To encode the time information from this daylength model, the authors hypothesize that two clock neurons found in fruit flies represent modulation of the expected solar azimuth collectively. Specifically, these DN1pB corresponds to two orthogonal directions of the solar azimuth. Based on previous studies, they further speculate that two types of TuBu1 neurons store the expected solar azimuth in the form of sin and cos (90deg phase offset). Finally, the solar declination and geometric latitude were modelled separately to account for season and location. Putting all these together, the authors produced a model that can track an absolute direction based on solar azimuth independent of time and place. This model was put to test in two simulation experiments to demonstrate the accuracy with simple daytime compensation and with complete model compensation. If the authors feel that the above summary is incorrect somewhere, it would be good to clarify that somewhere, as this is my understanding from reading the paper.

In general, this paper provides a comprehensive theoretical derivation on how time-compensated celestial navigation could work in insects. However, due to some knowledge gaps from the current literature, several mechanistic speculations had to be made in both the exact roles of certain neurons and the way solar azimuth is represented in the neurons.

- 1. As a reader, I would like to see more demonstration on the "alternative hypotheses" of these speculations. For instance, encoding a variable with orthogonal dimensions is convenient computationally but may not be how biological systems work. For TuBu1a and TuBu1b, is the 90deg offset necessary for the circuit to work or can they be in any known offset? Is there an alternative configuration for Fig 1h?*

Answer: We agree that it is very interesting to consider alternative hypotheses for the neural implementation of the functions we describe. Indeed, orthogonal dimensions are necessary for this model to work, but we believe there is accumulating evidence that this is a common feature in how the insect brain encodes information, particularly in pathways related to navigation. However, there are multiple ways to implement orthogonality, and for this specific example, the 90-degree offset in bumps could be obtained either as we suggest (in different neuron types, TuBu1a and TuBu1b) or between brain hemispheres, or through shifts in the downstream connections. We have added text in the discussion and a figure (Fig. 7) to highlight these alternative hypotheses:

Lines 506-515: "We note that the same trigonometric operations could be obtained by several alternative circuit configurations that are functionally equivalent to our model. For example, the cosine and sine of the solar azimuth could be encoded in left and right brain hemispheres (Fig. 7a) or implemented by offset connectivity (Fig. 7b). The number of ER5m and TuBu1 neurons could differ (Fig. 7c), as has been observed

in the *D. melanogaster* brain (Hulse et al, 2021), and the relative phase of their sinusoids be offset by an arbitrary amount (Fig. 7d).”

Fig. 7: Alternative circuits for the spatiotemporal integration. In our model, eight pairs of TuBu1a and TuBu1b neurons encode the sine and cosine of the sun’s azimuth from the animal’s heading (spatial representation) and single DN1pBE and DN1pBN neurons encode the sun’s azimuth from North (temporal representation). Each pair of TuBu1 neurons (modulated by a DN1pB neuron) targets an ER4m neuron and implements the required trigonometric identity. There are several alternatives to this implementation. (a) A single type of TuBu1 neurons could encode the sine and cosine of the sun’s azimuth in the left and right brain hemispheres respectively. (b) Two TuBu1 populations could have aligned activity bumps but express the sine and cosine of this angle through a 90° offset in their downstream connectivity. (c) Any relative population size of TuBu1 and ER4m could implement the same trigonometric identity as long as their connectivity is adjusted accordingly. (d) A homogeneous offset of the sinusoids by an arbitrary angle (θ) in TuBu1 and DN1pB does not affect the resulting ER4m sinusoid. Heterogeneous offsets of space (θ) in TuBu1 and time ($\theta' = \theta$) in DN1pB produce a constant offset in ER4m, maintaining a consistent geocentric encoding.

2. The central-place foraging validation is good. I understand that the agent performs a random walk initially in each trial, so the navigation circuit is challenged by a different homing initial condition (and directional integration). It would be useful to mark the accuracy as measured experimentally in some foraging insects on the figure for reference.

Answer: Although it is difficult to obtain comparable data (i.e., where it is clear that no other factors than celestial compass accuracy can be influencing navigation accuracy) we have updated Fig. 3 (now Fig. 5) to include the most relevant estimate, from desert ants navigating over comparable distances after displacement in relatively empty environments, of their accuracy in homing (Huber and Knaden, 2015). Interestingly, additional data suggests that outward paths (to a remembered food source) are more accurate than homeward paths (Pfeffer et al., 2015), which corresponds to our simulation results for the complete, but not the hour-angle, model. We have added discussion of this observation to the text describing these results.

Lines 338-344: “Experiments with desert ants suggest that their homing error is slightly higher than the one our simulations produced using any model (Huber and Knaden, 2015) (Fig 5d, green dashed line). Interestingly, the foraging routes of the real animals appear to be significantly more accurate than their homing routes (Pfeffer et al., 2015), which is only in line with the predictions of the complete model.”

Fig. 5: Computer simulated central-place foraging routes. At dawn, the simulated insect searches for food and stores the food location after finding it. It then returns to its nest. It tries to repeat foraging to the food location every hour until sunset, using (a) a model without time compensation, (b) the hour-angle model, or (c) our complete model. (d) Euclidean distance (m) of the search centroid from the feeder (red) or the nest (green) using the three models. The sample size in each box is $n = 9$. The green dashed line indicates the homing error of desert ants in a similar foraging scenario [38]. (e) Homing error as a function of foraging duration (normalised for foraging distance). Solid lines show the mean error when using the no-compensation (grey), the hour-angle (blue) or the complete model (yellow). Shaded areas are the 95 % confidence interval (CI). In all the results there is 20 % added compass noise.

3. For the migration experiment, it seems like the agent starts from the same location every time, so the only variation is the 20% added compass noise. This is likely not a good enough representation of natural variation. Perhaps the authors should consider initializing the

migration with at different time of the day, season, or minor changes in location. It would be interesting to know if the navigation model performs better at any time or location.

Answer: This suggestion to explore the effects of variation in start location, time of day and season was very helpful to enhance the analysis of our model performance. We kept the 20% compass noise and varied our experiments by randomly altering the starting location ($\pm 5^\circ$ in latitude and longitude) and time (± 72 hours) among runs. We also initialised the migrations from different locations along the Canadian border (butterflies), east African coast (dragonflies) and southeast Australia, allowing comparisons among different angles of travel. We have augmented our Fig. 4 (now Fig. 6) and text to include and comment these results.

Lines 395-429: “Figure 6a mimics the southward autumn migration routes of monarch butterflies using the hour angle (blue) and complete models (yellow), starting in late August from Mackinac Island (Michigan) and ending in late October in Michoacan (Mexico). Figure 6b shows similar migrations, starting from different locations across the Canadian border and California. Figure 6c summarises the performance of the two models in the above migrations. The variability in the performance comes from variations in the departure location ($\pm 5^\circ$ in longitude and latitude) and time (± 72 h). Overall, the performance of the two models seems very similar.

Figure 6d shows the simulated routes of Bogong moths during autumn migration from Montrose (Australia) south to Mount Bogong. Figure 6e shows similar migrations starting from different locations in Australia. As this migration occurs in the south hemisphere, we use the CCW version of the hour angle model. Figure 6f summarises the migration performance of the two models in the southern hemisphere, showing insignificant differences between the two models. The lower final distances from the goal than the butterflies are simply because the overall migration distances are shorter so less error accumulates.

Finally, Fig. 6g shows the simulated routes of the globe skimmer dragonflies during their spring migration from Mbekenyara (Tanzania) to Madirai (India), which is the world's longest transoceanic migration (approximately 4859 km). Figure 6h shows similar migrations starting along the east African coast. In these simulations, we use the equator-specific hour angle model. Figure 6i summarises the migration performance using the two models, where we see some more significant differences in their performance. Supplementary Fig. S5, S6 and S7 show the respective migrations of butterflies, moths and dragonflies using alternative hour angle models.”

Fig. 6: Computer simulations of insect migrations. (a) Example simulation of a monarch butterfly (*Danaus plexippus*) during its autumn migration, using a clockwise (CW) hour-angle model (blue) and complete model (yellow). Haversine distance: 3303.01 km. The red dashed arrow illustrates the straight line between the start and goal locations. (b) Examples of similar migrations that start from different locations along the Canadian border and California. (c) Haversine distance (km) between the butterflies' search centroid and the migration's destination point for each model and starting location. Variability comes from spatial ($\pm 5^\circ$) and temporal (± 72 h) variations at the beginning of the migration. (d) Similar example simulation for the Bogong moth (*Agrotis infusa*) during its autumn migration, using a counter-clockwise (CCW) hour angle and complete models. Haversine distance: 1114.65 km. (e) Other examples of the same migration, starting from different locations in Australia. (f) Haversine distance between the moths' search centroid and the migration's target location (similar to c, but for the moths). (g) Example simulation of a globe skimmer dragonfly (*Pantala flavescens*) during its spring migration, using the horizontal (east-west) component of the hour-angle model (vertical is set to zero) and the complete model. Haversine distance: 4859.20 km. (h) Similar simulations start from different locations on the east African coast. (i) The Haversine distance (km) of the dragonflies' search centroid from the target migration's target location (similar to c and f, but for the dragonflies). (j) Sensitivity analysis of the complete model to disturbances in estimating the sun's azimuth (grey; dashed: noise in the one-dimensional angle, solid: noise in the eight-dimensional neural representation of the angle), hour angle (blue) and magnetic inclination (yellow), pooled across the different migrations. In c, f and i the sample size is $n = 10$; in j it is $n = 15$ per 10% noise. In a-i, there is 20% added noise in both sensing and processing.

We also compared the performance of our models in the different seasons (± 3 months from original spring and autumn seasons). We created Supplementary Fig. S8 to show the performance of the different models in different seasons and commented on it in the text.

Lines 435-445: “The hour-angle model deviates more from the desired destination point for the dragonflies crossing the equator (Fig. 6i), but it can still drive the animal fairly close to its destination. The observed error results from a consistent bias rather than lower precision (Supplementary Fig. S8), which a constant offset could compensate for. Therefore, the only clear advantage of the complete model is that it allows full adaptation to different environments. In contrast, we had to choose the appropriate variant of the hour angle model for simulated migration above, below or across the equator.”

Supplementary Fig. S8: (a) Simulation of monarch butterflies (*Danaus plexippus*) during migrations in different seasons, using the clockwise hour-angle (blue) and complete models (yellow). The red dashed arrow illustrates the straight line between the start and destination points. Migrations in autumn and winter start from the Canadian border or California and end in Mexico. Migrations are reversed for spring and summer. (b) Similar simulation for Bogong moths (*Agrotis infusa*). Migrations in summer and autumn start from several places in Australia and end on Mount Bogong. Migrations are reversed for winter and spring. (c) Similar simulation of a globe skimmer dragonfly (*Pantala flavescens*). Migrations in spring and summer start from the eastern African coast and end in India. Migrations are reversed for autumn and winter. (d) Haversine distance (km) of the search centroid from the desired destination of butterflies, conditioned to season and model. Similar Haversine distance for (e) the moths and (f) dragonflies. For each animal, the sample size is $n = 10$, with random variations to the starting location ($\pm 5^\circ$) and time (± 72 h). There is also 20% added compass and processing noise in all results.

4. If the sole objective was to show that equator crossing requires complete compensation, does it require a simulation as the sign flip is so drastic in the model? My suggestion would be to

explore how to add value to the migration simulations. For instance, is the direction variation in the simulation consistent with experimentally measured migration paths, or direction choices in navigating insects. It would be informative to build a simple phenomenological model of navigation based on experimental data and compare to this mechanistic model of navigation. It would be also informative to see under what other conditions the model might fail (if any).

Answer: We appreciate this suggestion but feel it is beyond the scope of our work to fully investigate mechanisms of migration, e.g., by constructing phenomenological models fitted to experimental data (which in any case is limited in respect of providing detailed migration paths, or accuracy of individuals). Our intent in this paper is rather to explore the consequences, in migrating insects, of our proposed mechanisms for time compensation in the use of a celestial compass. We updated our text to clarify this.

Lines 367-376: *“We therefore simulated insect migration, over realistic travel distances and directions, using migration routes inspired by three different insect species: the monarch butterfly *D. plexippus*, the Bogong moth *Agrotis infusa*, and globe skimmer dragonfly *Pantala flavescens*. We selected these species to demonstrate migration in different characteristic locations on earth relative to its equator: above (*D. plexippus*), below (*A. infusa*), or across (*P. flavescens*). In this way, we test our model in locations where the sun moves CW, CCW, or switches its moving pattern during the migration.”*

We have also updated our model for the equator-crossing migration to test whether the ‘sign flip’ could be handled by an hour-angle model that sets the north-most axis to zero, and thus tracks only the switch of the sun azimuth from east to west (neglecting whether this is clockwise or counterclockwise).

Lines 130-135: *“Near the equator, the sun moves almost vertically (crossing the zenith) from east to west. In this case, a zero north-most component ($\omega_N = 0$; no vertical vector in Fig. 1d) would be more accurate and avoids any assumption regarding CW or CCW sun movement.”*

We find that, although not as accurate as the complete model compensation, this does allow the migration to reach the approximate intended location. Correspondingly we now believe the complete model may not be necessary even in this case, although it would remain necessary for an insect that needed to flexibly adapt to different geographical locations. We have updated the results and discussion to reflect this insight. Combined with our additional migration simulations (see Fig. 6, Supplementary Figures S5, S6, S7, and S8) we hope that this work provides a solid basis for future exploration of insect migration.

Lines 335-345: *“The hour-angle model deviates more from the desired destination point for the dragonflies crossing the equator (Fig. 6i), but it can still drive the animal fairly close to its destination. The observed error results from a consistent bias rather than lower precision (Supplementary Fig. S8), which a constant offset could compensate for. Therefore, the only clear advantage of the complete model is that it allows full adaptation to different environments. In contrast, we had to choose the appropriate variant of the hour angle model for simulated migration above, below or across the equator.”*

Overall, this is a very nice paper and would be informative to the community.

Answer: We thank the reviewer for their constructive feedback, which has certainly improved our paper.

Reviewer #2 (Remarks to the Author):

This manuscript describes a speculative model by which insects navigating via celestial cues might compensate for the movement of the sun during the day. The authors have built a navigation model that combines input from solar position, timekeeping neurons, and magnetic fields. They develop a simple model based on the hour angle of the sun, and more complete model that adds an estimate of latitude acquired via geomagnetic inclination. They then simulate place foraging over periods of hours, and distant migrations that last over months, to demonstrate approximately successful navigation.

To their credit, the authors have worked to keep the model based on input and circuits that real insects are known to use, such as time clock neurons and magnetic inclination. Every step seems plausible, at least in some insects, and aimed at explaining behavior that is observed, but not fully understood. To this end the models do offer believable mechanisms by which solar time compensation could be achieved in insects.

Answer: We are grateful for the reviewer's comments below, which we believe strengthen our manuscript.

1. *My main concern is that this process requires many steps of neural calculations that aren't directly tested. I have little sense of how likely this particular model is, and for which insects.*

Answer: We appreciate this concern but would emphasise that the key elements presented here actually involve very few steps of neural calculations. Indeed, the core idea consists of essentially one step, the combination of (known) time encoding with (known) spatial encoding, based on identified circuits in the insect brain. The spatial encoding is well supported by previous (biologically grounded) modelling, as are all the downstream neural processes to support the navigation behaviour. We agree that the input to the temporal encoding is much more speculative, but believe it is an important contribution to have identified some crucial properties this encoding needs to satisfy, such as being able to synchronise to daylength, not just sunrise. With regard to how temporal and spatial encoding is combined, we believe this is the first attempt to describe a concrete neural mechanism to provide the required function, and in response to this comment and related comment 1 from reviewer 1, we have added a figure (Fig. 7) and further discussion of how alternative neural implementations might support the equivalent function.

Lines 506-515: *"We note that the same trigonometric operations could be obtained by several alternative circuit configurations that are functionally equivalent to our model. For example, the cosine and sine of the solar azimuth could be encoded in left and right brain hemispheres (Fig. 7a) or implemented by offset connectivity (Fig. 7b). The number of ER5m and TuBu1 neurons could differ (Fig. 7c), as has been observed in the *D. melanogaster* brain (Hulse et al, 2021), and the relative phase of their sinusoids be offset by an arbitrary amount (Fig. 7d)."*

Fig. 7: Alternative circuits for the spatiotemporal integration. In our model, eight pairs of TuBu1a and TuBu1b neurons encode the sine and cosine of the sun's azimuth from the animal's heading (spatial representation) and single DN1pBE and DN1pBN neurons encode the sun's azimuth from North (temporal representation). Each pair of TuBu1 neurons (modulated by a DN1pB neuron) targets an ER4m neuron and implements the required trigonometric identity. There are several alternatives to this implementation. (a) A single type of TuBu1 neurons could encode the sine and cosine of the sun's azimuth in the left and right brain hemispheres respectively. (b) Two TuBu1 populations could have aligned activity bumps but express the sine and cosine of this angle through a 90° offset in their downstream connectivity. (c) Any relative population size of TuBu1 and ER4m could implement the same trigonometric identity as long as their connectivity is adjusted accordingly. (d) A homogeneous offset of the sinusoids by an arbitrary angle (θ) in TuBu1 and DN1pB does not affect the resulting ER4m sinusoid. Heterogeneous offsets of space (θ) in TuBu1 and time ($\theta' = \theta$) in DN1pB produce a constant offset in ER4m, maintaining a consistent geocentric encoding.

2. Further, estimates of magnetic inclination, necessary for the full model, seem particularly tricky to me. The only citation given to support this is for the monarch butterfly, which curiously migrated fine in their simulations with just the hour-angle model. Does the globe skimmer dragonfly possess this sense? The authors seem to claim that it must, as it crosses the equator,

which required the full model. But magnetic inclination is especially low around the equator—how well could an insect estimate this variable, and how sensitive would it be to local or secular magnetic variation?

Answer: This is a very relevant comment, and as a consequence we have explored whether there is an alternative by which the dragonfly (or other equator-crossing) insects could support navigation without explicitly sensing the latitude (through magnetic inclination or other means). We realised that a variation on the hour-angle model that sets the north-most element to zero produces an estimate of azimuth that simply switches from directly east to directly west at solar noon, which adequately approximates the path of the sun near the equator, neglecting any assumption of clockwise or counterclockwise rotation. Using this model for the dragonfly produced relatively successful migration paths, although not as accurate as the complete model. We have added these results and appropriate discussion to the text. We have also carried out a sensitivity analysis including noise in the magnetic inclination as detailed in our response to the next comment.

Lines 130-135: *“Near the equator, the sun moves almost vertically (crossing the zenith) from east to west. In this case, a zero north-most component ($w_N = 0$; no vertical vector in Fig. 1d) would be more accurate and avoids any assumption regarding CW or CCW sun movement.”*

Lines 430-445: *“The complete model generally resulted in straighter migration routes than the hour-angle models. However, the deviations induced by the less accurate hour-angle model largely cancel out during the day. As a result, in the single-hemisphere migrations, both models could bring the animals close to their destinations (Fig. 6c and f). The hour-angle model deviates more from the desired destination point for the dragonflies crossing the equator (Fig. 6i), but it can still drive the animal fairly close to its destination. The observed error results from a consistent bias rather than lower precision (Supplementary Fig. 8), which a constant offset could compensate for. Therefore, the only clear advantage of the complete model is that it allows full adaptation to different environments. In contrast, we had to choose the appropriate variant of the hour angle model for simulated migration above, below or across the equator.”*

3. *Other than the comparison of complete and hour-angle models, the model lacks a rigorous sensitivity analysis. Many inputs, such as sun position, time-estimate of sunrise, and magnetic field inclination, are certainly subject to error (such as by clouds). How much does this matter?*

Answer: We have added a sensitivity analysis which models error in sun position, time estimation and magnetic field inclination as random uniformly distributed noise of different magnitudes. We find that noise of up to 25% of the maximum possible measurement value in magnetic inclination (i.e. up to +/-45 degrees, as above) and time (i.e. up to 6 hours) does not significantly affect accuracy; noise in the sun's position estimations has smaller effect, because it is encoded by a larger population of neurons (eight) than the magnetic inclination and time information (two each). This latter point was verified by running the same analysis when noise was introduced in the sun azimuth itself instead of its neural representation, which led to a more rapid decrease in navigation accuracy as the noise level increased. We added these results in Fig. 6j and updated the text accordingly:

Lines 446-462: *“We performed further sensitivity analysis by introducing random sensor noise from a uniform distribution, with the upper and lower bounds set as a percentage of the maximum possible measurement value (Fig. 6j). For the hour angle, the error (average distance from goal across all migrating species) becomes significant only for unrealistic levels of noise in estimating the time (more than 25% noise, corresponding to 6 h). Magnetic inclination measurements are expected to be noisier (especially near the equator) but had to exceed 45° to produce noticeable error. If noise in the sun position is introduced independently in each TuBu1 neuron, then its effect averages out across the eight-dimensional encoding of the angle, leading to little change in migration error as the noise level is increased (solid grey*

line in 6j). If the noise is added to the sun's angle before TuBu1, the error increases more rapidly, from around 10 ° (dashed grey line in Fig. 6j).”

Fig. 6j: Sensitivity analysis of the complete model to disturbances in estimating the sun's azimuth (grey; dashed: noise in the one-dimensional angle, solid: noise in the eight-dimensional neural representation of the angle), hour angle (blue) and magnetic inclination (yellow), pooled across the different migrations. The sample size is $n = 15$ per 10% noise.

4. *My concern is that a complex model with many free parameters and little ground truthing might need much revision as our specific knowledge improves.*

Answer: Although it might appear that the model has many free parameters, e.g., in connection weights between neurons, in fact these parameters are strongly constrained by the trigonometric principles that the network needs to implement. In practice, there are very few (only 3) parameters that were optimised for the model using realistic constraints. Nevertheless, we agree the model might need much revision as our specific knowledge improves. In our view, this does not diminish the usefulness of publishing the model in its present form to help guide future investigations.

Minor:

5. 280: *"This describes an oscillation with an annual period, which can be further decomposed into two sinusoidal functions (Fig. 2b)."*

This isn't what I normally think of as "decomposing" a sinusoid, since the cosine of this sinusoid has double its frequency. In other words it is not in terms of simpler functions. But further, the motivation for this step isn't clear to me, other than that it yields functions you want to use later?

Answer: We understand the confusion of the reviewer and we changed our wording to clarify this. Although the solar declination is an annual oscillation, it is also an angle. If we think of it as a unit vector in polar coordinates, the sine and cosine of this vector can represent this angle, while their combination can reconstruct it. We hope our wording clarifies this:

Lines 259-261: *"This oscillating angle with an annual period can be described by its sine and cosine components (Fig. 4b)."*

We also updated our Fig. 2c (now Fig. 4c) to make the geometrical relation of the different components more apparent.

Fig. 4: Complete model of the sun's movement. (a) The solar declination is the earth's latitude where the sun is exactly at the zenith, which is equivalent to the angle of the line connecting the centres of the earth and sun from earth's equator. (b) The solar declination (δ) is represented by a sinusoidal function of time with an annual period. This can be encoded in a two-dimensional direction as described by its sine (δ_n) and cosine (δ_o). **Vertical axis in arbitrary units.** (c) A schematic showing how the solar declination adds to the north-most component of the hour angle, influencing both the sunrise (sr) time and solar azimuth (α) based on the season. The horizontal and vertical axes are the sine (east-most) and cosine (north-most component) of the hour angle. (d) The geomagnetic inclination (μ) is the angle between the geomagnetic field and the earth's surface. (e) The geomagnetic inclination is a monotonic function of the geometric latitude (ϕ), which can also be described by its sine (ϕ_n) and cosine (ϕ_o), representing a two-dimensional direction. **Vertical axis in arbitrary units.** (f) A schematic showing how the latitude multiplies with the north-most component of the hour-angle. It influences its magnitude and flips it in the south hemisphere, allowing for clockwise (CW) and counter-clockwise (CCW) movements of the sun. Axes similar to c. (g) The full solar azimuth model. The proposed circuit combines information from the geomagnetic inclination and daily and annual clocks, to accurately estimate the sun's course during the day. This estimate is a vector with a north-most (α_N) and east-most (α_E) component.

Regarding the motivation for this step, the reviewer's intuition is correct: it yields functions we want to use later. In our models, the key components are neurons whose activity must be bounded. Representing an angle directly as strength of neural activity is a poor computational solution (e.g. not easily expressing that $360 = 0$) and does not reflect our knowledge of how neural activity represents angles in the insect brain. Expressing the angle by its sine and cosine creates convenient boundaries for the activity and aligns with our claim that neural activity and connections in navigation-related areas of the insect brain must conform to trigonometry.

Lines 476-479: “We suggest that this correction relies on trigonometric principles for spatial and temporal processing, and show a plausible neural mechanism by which the required trigonometry could be implemented.”

6. Fig 2

Many axes are not labeled on the figure, or adequately described in the legend or main text. For example, in 2c it is quite difficult to determine what is plotted when units are missing from both horizontal and vertical axes.

Answer: We are sorry this figure (now Fig. 4, as shown above) caused confusion. We have revised the figure, replacing the previous schematic illustrations (c and f) with geometric explanations to describe the role of solar declination and latitude in the transformation of the hour angle to the solar azimuth estimate. Panels **b** and **e** are unchanged as they already had labelled axes (noting that for the sinusoids (lower plots), the y-axis lacks units as these are arbitrary). We hope the figure is now clear.

7. 430 “Finally, Figure 4e-g shows the simulated routes of the globe skimmer dragonflies during their spring migration from Mbekenyara (Tanzania) to Madirai (India), which is the world’s longest transoceanic migration (approximately 4859 km).”

I don’t understand, did this simulated migration also include “daily travel capacity of insects is eight hours (at 2.5 m sec^{-1} , based on the speed of *D. plexippus*), and they need one-hour breaks every hour for feeding or rest”, even though those aren’t part of this migration?

Answer: In our simulations, all insects have the same daily migrating pattern, based on the monarch butterfly. This was for simplicity, as it was not our aim to model migration in detail, but rather to assess the proposed mechanisms of time compensation over realistic geographic ranges. Rerunning the simulation without the one-hour breaks produced smoother paths but did not change the overall results. We have now clarified this in the text by adding:

Lines 338-392: “The daily travel capacity was set at eight hours (at 2.5 m sec^{-1} , based on the speed of *D. plexippus*), with one-hour breaks every hour for feeding or rest (using continuous travel produced somewhat smoother courses but did not change the overall results).”

Lines 802-809: “For the migration experiments, we assume that insects can travel for a maximum of 8 h per day (only between sunrise and sunset), they need to stop every hour for rest and feeding, and that their stops last for 1 h. The speed of the insects was set to $v(t) = 2.5 \text{ m sec}^{-1}$, which is based on the speed of *D. plexippus* monarch butterflies (although this could differ in reality between insects), and the time step used was $dt = 50 \text{ min}$.”

Manuscript NCOMMS-24-42339

Response to reviewers

Reviewer #2 (Remarks to the Author):

The authors have done an excellent job of addressing my comments, and I'm especially pleased with the addition of a sensitivity analysis. My only remaining point of confusion is with my final comment, where the authors seemed to simulate 8 hour daily travel and hourly rest periods for the globe skimmer migration between East Africa and India. They now comment that continuous travel makes little difference to the results of the model, and skipping the rest period produced no significant changes. But these details are just inserted without specific reference to the globe skimmers. The reason for my initial confusion was that I was unsure if these parameters were the same for the globe skimmer migration, so I think it would be most helpful to mention that specifically when discussing the dragonfly.

Answer: We have altered our text to clarify this in the paragraph where we discuss the dragonfly results.

p. 5, §4, line 3: *“Although dragonflies may fly more continuously during their transoceanic migration, for consistency we used the same migration time pattern as for the monarch butterfly (travel at 2.5 m sec^{-1} with foraging brakes). Exploratory simulations using continuous migrations produced smoother routes but led to the same conclusions.”*